# A robust, semi-automated approach for counting cementum increments imaged with synchrotron X-ray computed tomography

Elis Newham [1,2]*, Pamela G. Gill[3,4], Kate Robson Brown[5,6], Neil J. Gostling[7], Ian J. Corfe[8,9], Philipp Schneider [1,10]*

1 Bioengineering Science Research Group, Faculty of Engineering and Physical Sciences, University of Southampton, Southampton, United Kingdom, 2 School of Engineering and Materials Science, Queen Mary University of London, London, United Kingdom, 3 School of Earth Sciences, University of Bristol, Bristol, United Kingdom, 4 Department of Earth Sciences, Natural History Museum, London, United Kingdom, 5 Department of Mechanical Engineering, University of Bristol, Bristol, United Kingdom, 6 Department of Anthropology and Archaeology, University of Bristol, Bristol, United Kingdom, 7 School of Biological Sciences, Faculty of Environmental and Life Sciences, University of Southampton, Southampton, United Kingdom, 8 Developmental Biology Program, Institute of Biotechnology, University of Helsinki, Helsinki, Finland, 9 Geological Survey of Finland, Espoo, Finland, 10 High-Performance Vision Systems, Center for Vision, Automation & Control, AIT Austrian Institute of Technology, Vienna, Austria

* elis.newham@googlemail.com (EN); p.schneider@soton.ac.uk (PS)

**Data Availability Statement:** All data supporting this study will be made openly available from the

## Abstract

Cementum, the tissue attaching mammal tooth roots to the periodontal ligament, grows appositionally throughout life, displaying a series of circum-annual incremental features. These have been studied for decades as a direct record of chronological lifespan. The majority of previous studies on cementum have used traditional thin-section histological methods to image and analyse increments. However, several caveats have been raised in terms of studying cementum increments in thin-sections. Firstly, the limited number of thin-sections and the two-dimensional perspective they impart provide an incomplete interpretation of cementum structure, and studies often struggle or fail to overcome complications in increment patterns that complicate or inhibit increment counting. Increments have been repeatedly shown to both split and coalesce, creating accessory increments that can bias increment counts. Secondly, identification and counting of cementum increments using human vision is subjective, and it has led to inaccurate readings in several experiments studying individuals of known age. Here, we have attempted to optimise a recently introduced imaging modality for cementum imaging; X-ray propagation-based phase-contrast imaging (PPCI). X-ray PPCI was performed for a sample of rhesus macaque (*Macaca mulatta*) lower first molars (n = 10) from a laboratory population of known age. PPCI allowed the qualitative identification of primary/annual versus intermittent secondary increments formed by splitting/coalescence. A new method for semi-automatic increment counting was then integrated into a purpose-built software package for studying cementum increments, to count increments in regions with minimal complications. Qualitative comparison with data from conventional cementochronology, based on histological examination of tissue thin-sections, confirmed that X-ray PPCI reliably and non-destructively records cementum increments (given the appropriate preparation of specimens prior to X-ray imaging). Validation of

University of Southampton repository ("https://doi.org/10.5258/SOTON/D1722").

**Funding:** This study was part-funded by a Natural Environmental Research Council/Engineering and Physical Sciences Research Council doctoral candidateship (UK; grant number NE/R009783/1). Funding was also provided by Ginko Investments Ltd (Bristol, UK), and the Academy of Finland.

**Competing interests:** The authors have read the journal's policy and have the following competing interests: The authors received material support in the form of travel to the Swiss Light Source Synchrotron for our experiment, consisting of airfare, guesthouse reservations, and external hard drives. This does not alter our adherence to PLOS ONE policies on sharing data and materials. Travel to the Swiss Light Source Synchrotron for our experiment, consisting of airfare, guesthouse reservations, and external hard drives.

the increment counting algorithm suggests that it is robust and provides accurate estimates of increment counts. In summary, we show that our new increment counting method has the potential to overcome caveats of conventional cementochronology approaches, when used to analyse three-dimensional images provided by X-ray PPCI.

# 1. Introduction

Mammalian teeth comprise three principal mineralised tissues: enamel, dentine and cementum. Each tissue provides its own record of growth via incremental patterns observed using thin-section histology that track periodic changes in their growth rates. Development of enamel and dentine is largely truncated after the maturation of the tooth. However, cementum grows continuously throughout life and provides a more complete record of an individual's life history [1]. Incremental features found in cementum are well understood to have a circum-annual periodicity, with one thick translucent increment, and one thin opaque increment formed every year (when viewed in thin-section under transmitted light microscopy; Fig 1). These contrasting opacities are hypothesised to be primarily due to seasonal changes in the mineralisation rates of the cementum hydroxyapatite matrix. Cementum is composed of bundles of collagen fibres emanating from both the cementum itself (intrinsic fibres), or from the periodontal ligament or PDL (extrinsic 'Sharpey's' fibres), wrapped within a hydroxyapatite matrix [2]. Unfavourable growth seasons promote a reduction in the deposition rate of this matrix, while the mineralisation rate is unaffected [3]. This produces ultra-mineralisation of thinner portions of cementum during these periods ([1, 4]; though see [3] and [5] for mineral distribution mapping studies that suggest higher mineral densities in wider, not narrower, cementum increments).

The proportion of collagen fibres emanating from either the cementum or the PDL defines the two major cementum tissue types. Acellular extrinsic fibre cementum (AEFC) contains predominantly Sharpey's fibres from the PDL, while cellular intrinsic fibre cementum (CIFC) contains only fibres originating from the cementum itself. While the distribution of either tissue is not homogeneous, there is consensus that AEFC is concentrated in the crownward/coronal third of the root, while CIFC is the predominant tissue of the apical third of the root [6]. AEFC has the most regular periodicity and provides the most consistent growth record in the cementum. CIFC grows more sporadically than AEFC with a less precise periodicity and is known to nucleate within regions of the tooth that experience anomalously high occlusal forces. As such, AEFC is usually the recommended cementum tissue for cementochronological studies and the tissue focussed on for this study ([6] and references within).

Several studies have questioned the accuracy, precision, and reliability of current methods of imaging and analysing cementum increments [7, 8]. CIFC cementum is a dynamic, biomechanically responsive tissue and increments are known to both split and coalesce. These phenomena are also known to occur in AEFC, which can undermine confidence in their counts as an estimate of age at death. The overwhelming majority of previous studies have relied on thin-sectioning samples to image increments using light microscopy [6]. This allows increments to be viewed at high spatial resolutions, and offers a range of optical [9] and digital image processing [10, 11] methods that filter and highlight increments to varying degrees. However, the destructive nature of thin-sectioning and the restrictive two-dimensional (2D) perspective offered by histological sections limit the understanding of the complex nature of cementum and its increments [7]. There is a wide range in the reported accuracy (the

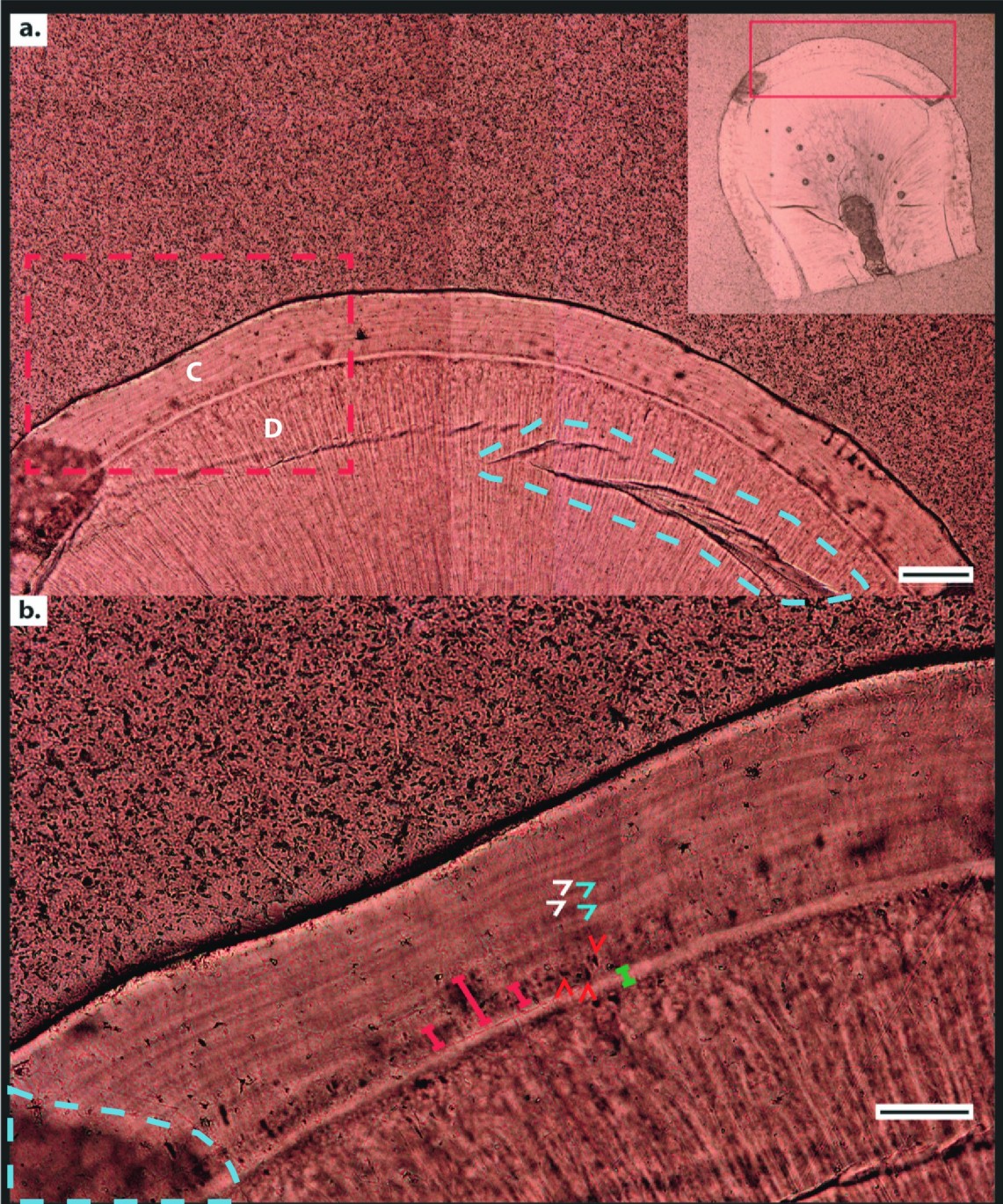

**Fig 1. Histological data of rhesus macaque (*Macaca mulatta*) cementum in the permanent lower first molar tooth.** (**a**) 70 μm-thick transverse reflected light digital micrograph of the cementum tissue from the coronal third of the tooth root at 20× magnification. Cementum (C) is defined as the tissue wrapping around the circumference of the dentine (D), comprising a series of circumferential increments. Blue dashed line highlights a surface scratch created during the thin-sectioning process. (**b**) Detail marked by red dashed box in (**a**) displaying cementum increments in acellular extrinsic fibre cementum (AEFC), and cellular voids left by cementoblasts in cellular intrinsic fibre cementum (CIFC) under higher resolution (50× magnification). White arrowheads highlight 'light' increments, blue arrows highlight 'dark' increments, and red arrows highlight cementoblasts. Red bracketed lines highlight intrinsic fibre cementum (CIFC). Green bracketed line highlights the hyaline layer of Hopewell-Smith, demarcating the cemento-dentine boundary. Scale bar in (**a**) represents 100 μm. Scale bar in (**b**) represents 30 μm.

proximity of estimated increment counts to known/true chronological age in years) and precision (repeatability and reproducibility between researchers) of increment counts [7, 12, 13], despite generally high reported accuracy and precision across multiple studies [6].

Computer vision and image processing have been explored to aid human counting of cementum increments, and to overcome the need for human counting itself [14, 15]. Peaks and troughs in cementum opacity or greyscale 'luminance' can be extracted along radial transects through the cementum from digital micrographs of thin-sections using image processing software and studied by using 'Digital Cementum Luminance Analysis' (DCLA) [11] (Fig 2A and 2B). Light increments are represented by distinct peaks in luminance values, and dark increments by distinct troughs in luminance values (Fig 2B). These patterns are interpreted by either manually counting peaks and/or troughs, or through peak/trough detection algorithms (e.g. 'Find peaks' in ImageJ/Fiji; [16]). This abstraction of increments to peaks and troughs offers a less subjective method for manually counting increments, compared to directly reading thin-section images (Fig 2B). Further, the use of numerical greyscale values to separate neighbouring light and dark increments allows quantitative thresholds to be defined for distinguishing individual increments. These thresholds represent a specific greyscale value from which peaks/troughs representing light/dark increments must differ from either the last trough or peak (respectively), or from the mean value of greyscale for the transect under study, in order to be identified as a 'genuine' increment.

However, DCLA methods have so far relied upon *a priori* assumptions regarding the contrast in greyscale values caused by incrementation versus those caused by other sources such as image noise. The chosen threshold value for distinguishing increments in each DCLA method is specific to the image technique and hence image data in the original study, and so may not be robustly applied to data from other imaging modalities or techniques, or to taxa that fail to meet the specified threshold specified in the original study. Thus, the next stage in DCLA development should focus on developing a more flexible strategy for distinguishing cementum increments based on relative instead of absolute greyscale distribution criteria.

Three-dimensional (3D) imaging, such as high-resolution computed tomography (CT) using X-ray propagation-based phase-contrast imaging (PPCI) at synchrotron radiation (SR) sources, may overcome the limitations posed by counting cementum increments in histological thin-sections. SR CT has revolutionised the study of other hard tissue microstructures such as vascular and cellular networks in bone [17–21] or growth increments in enamel [22–27]. SR CT imaging of such tissues provides a 3D context to the study of internal structures at sub-micrometre levels, and, with high signal-to-noise ratios (SNRs) and high contrast-to-noise ratios (CNRs), provides high levels of image quality. PPCI through SR CT has recently allowed increments to be followed through the cementum of the teeth of archaeological humans [28–30] and macaque monkeys and early mammal fossils [31, 32], overcoming the limitations of 2D thin-section-based imaging. The study of Le Cabec et al. [30] investigated a sample of human teeth from an archaeological population of known age at death, and reported high precision between counts performed by different observers, and for repeated counts performed by the same observer. However, although a strong correlation was found between age estimated by increment counts and known age, the accuracy (proximity of estimated age to known age) of estimates fell from 2.5 years in individuals 20–29 years old to 28 years in individuals 60–89 years old (with counts consistently underestimating actual age). Due to the uniqueness of their archaeological sample, the canines studied in Le Cabec et al. [30] could not be thin-sectioned. On this account, it could not be determined whether the nature of the source of this inaccuracy was biological, diagenetic (chemical changes to the cementum changing and overprinting original increments), or technical (insufficient image contrast between increments due to similar material properties in terms of X-ray interactions and/or due to

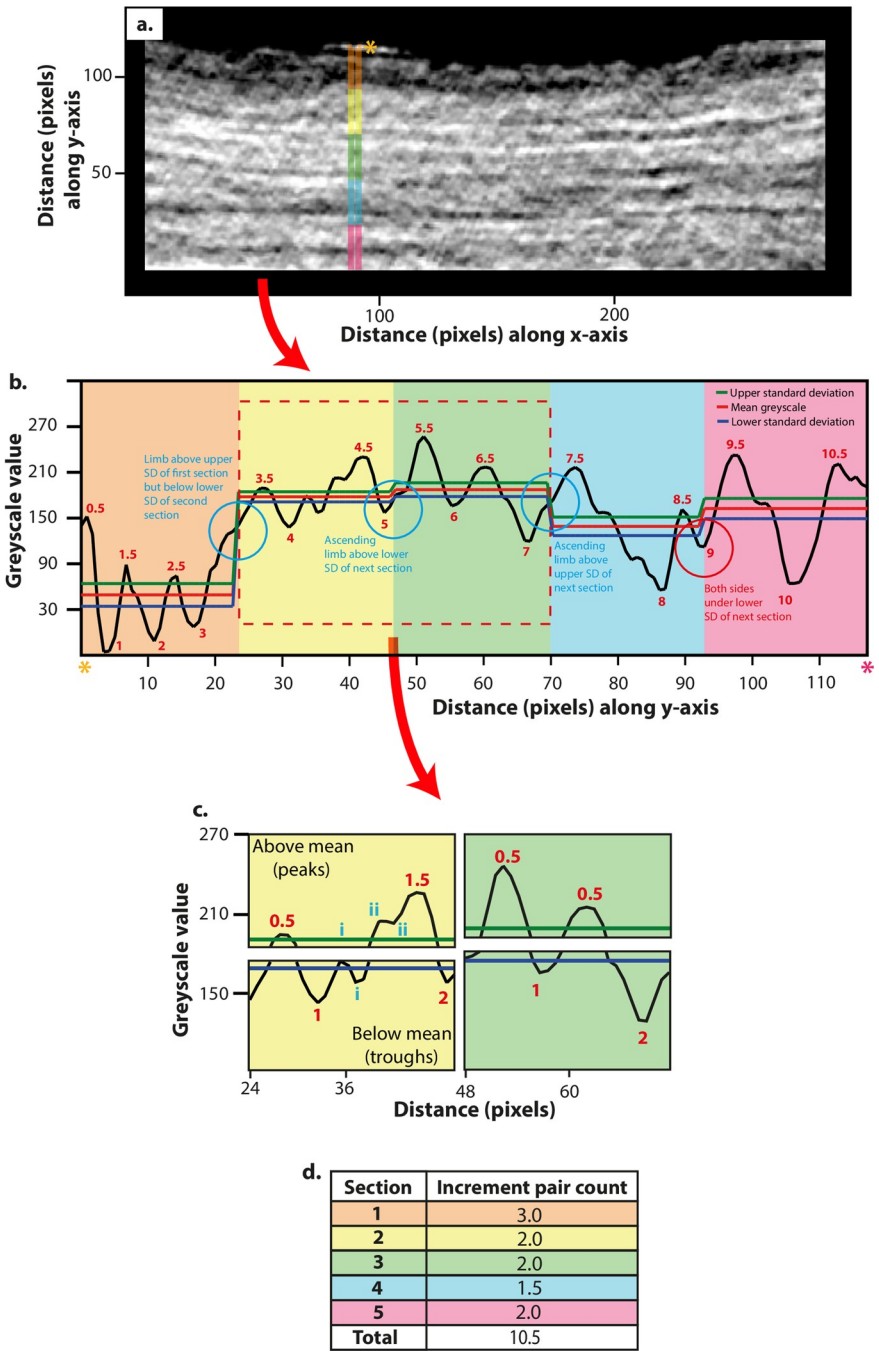

**Fig 2. Tomographic cementum increment counting.** (**a**) Straightened, filtered PPCI SR CT image of *Macaca mulatta* cementum (see Fig 5). (**b**) Plot of greyscale values along transect highlighted by the 10 pixel-thick coloured band in (**a**) with outer-cementum surface highlighted with orange asterisk and cemento-dentine boundary by red asterisk. Transects are split into five sections. Light/dark increment pairs are distinguished as peak/trough systems in greyscale values, where both the peak and trough depart from the mean greyscale value of that section (red line), beyond the half standard deviations of greyscale values within the section above the mean (green line) and below the mean (blue line), respectively. (**c**) Sections split into their upper and lower datasets comprising of peaks and troughs in greyscale that exceed beyond the upper and lower standard deviation (respectively). Here, peak/trough pairs are counted, denoted with red numbers. Troughs and peaks that are not counted are denoted in blue numerals, as they either do not exceed the standard deviation of that section or are less than three pixels away from the last respective peak/trough. (**d**) Resulting increment pair counts for each section as seen in (**b**).

unsuitable imaging and/or CT reconstruction settings). Further optimisation of cementum analysis using SR CT must be accompanied by comparison with histological data of the same regions of interest (ROIs), in order to relate image quality of SR CT data to data from thin-section histological imaging, the current 'gold standard' for cementum imaging.

Here, we aim to optimise PPCI through SR CT for studying cementum increments in 3D, in comparison with the respective histological data in 2D. We will particularly assess the potential for overcoming the limitations identified in current approaches for counting cementum increments. As a second objective, we provide a validated, semi-automated and robust algorithm for user-independent counting of cementum increments.

## 2. Materials and methods

### 2.1. Teeth samples

This study is focused on the analysis of the permanent right lower first molars (M1) from a sample of 10 female Rhesus macaques (*Macaca mulatta*), raised under laboratory conditions and bred for biomedical research at the Primate Breeding Facility of Public Health England, Salisbury (UK) (Table 1). All animals used here were routinely monitored and checked for primate-borne diseases of risk to humans (including hepatitis B, herpes B and tuberculosis) and were humanely killed using an overdose of pentobarbital, under Home Office establishment licence 70–1707, due to being unfit for breeding or whole-animal scientific procedural use. No animal was killed for the specific purpose of this study. Once the animals were killed, their lower jaws were mechanically dislocated and removed by Public Health England. Lower jaws were then freeze-stored at -20˚C prior to further tissue preparation. The studied sample was classed as Category B biological waste by the UK government. As no animal was sacrificed or harmed for the purpose of this study, no animal research ethics committee approval was needed. No permits were required for the described study, which complied with all relevant regulations under the UK Animals (Scientific Procedures) Act 1986 (ASPA).

To prepare specimens, lower jaws were first mechanically cleaned of soft tissue using surgical tools (scalpel, scissors and tweezers). The coronoid and angular processes were then removed using a handsaw. Once prepared, specimens were bathed in tap water in a sealed plastic container, which was stored in a fume cupboard for three weeks (21 days). This procedure was adopted to rot away the periodontal ligament and alveolar soft tissue that could

**Table 1. Life history data, known increment pair counts and estimated increment pair counts for each of the 10 female rhesus macaque (*Macaca mulatta*) individuals studied.**

| Specimen | Age (years) | DOB | DOD | Known increment pair count | | Estimated increment pair count | |
|---|---|---|---|---|---|---|---|
| | | | | Minimum | Maximum | Mean | Standard deviation |
| k49 | 12 | 09.04.03 | 08.04.15 | 10 | 11 | 10 | 0.95 |
| k91 | 11.5 | 06.10.03 | 10.04.15 | 9.5 | 10.5 | 10 | 0.81 |
| k23 | 12 | 09.03.03 | 09.04.15 | 10 | 11 | 10 | 0.89 |
| k24 | 12 | 12.03.03 | 08.04.15 | 10 | 11 | 10 | 0.96 |
| l10 | 11 | 20.02.04 | 08.04.15 | 9 | 10 | 9.5 | 0.94 |
| l14 | 11 | 26.02.04 | 10.04.15 | 9 | 10 | 9.75 | 0.86 |
| l56 | 11 | 14.04.04 | 10.04.15 | 9 | 10 | 10 | 0.92 |
| l59 | 11 | 17.04.04 | 09.04.15 | 9 | 10 | 9.5 | 0.71 |
| k16 | 10.5 | 16.09.04 | 09.03.15 | 8.5 | 9.5 | 9 | 0.52 |
| t46 | 5.5 | 11.03.10 | 07.07.15 | 3.5 | 4.5 | 5 | 0.94 |

DOB = date of birth; DOD = date of death. Mean increment pair counts are rounded to the nearest 0.25 years for comparison with known increment pair counts.

not be mechanically removed. After three weeks, teeth were sufficiently loose within the jaw to be easily removed using surgical pliers. The left and right permanent M1 teeth of all animals were fixed in 10% paraformaldehyde (PFA) solution for 10 days to minimise risk of infection.

Finally, to minimise the X-ray absorption by dense mineralised tissue that was not of interest (i.e. enamel, crown), the roots were mechanically separated before imaging. The crowns of all teeth were removed using a Buehler IsoMet® Low Speed precision sectioning saw equipped with an Acuthin^tm blade (Buehler Ltd, Lake Bluff, IL, USA). Using the same saw, the mesial and distal roots of each M1 tooth were mechanically separated and mounted on 2 mm-thick carbon fibre rods (CR200600; Ripmax Ltd, Enfield, UK) cut to 1.5 cm length, using cyanoacrylate superglue. While this is a destructive preparation technique (relative to Le Cabec et al. [30]), we determined it as necessary to provide the optimum sample dimensions for SR CT imaging. Following PPCI experiments, transverse histological thin-sections of approximately 70 μm thickness were prepared, cut through the region of ROIs that were acquired by PPCI through SR CT, following the method outlined in Newham et al. [31].

Chronological age at death in years for each individual is indicated in Table 1. This data point, and the average age of eruption of lower permanent M1 teeth for captive populations of *Macaca mulatta* (approximately 12–18 months; [33]) provided an expected increment count for each individual (age at death minus average age at eruption). However, a potential variation of six months for M1 eruption necessitated the use of a minimum expected increment count, and a maximum expected increment count (one increment higher than the minimum expected count) (Table 1). All thin-sections and remaining whole tooth samples were stored at the University of Southampton and are accessible upon request to EN and/or PS.

## 2.2. X-ray PPCI of cementum

PPCI for this study was performed during a three-day experiment at the TOMCAT beamline of the Swiss Light Source (SLS) (Experiment 20151391, 15–18 March 2016). The station at TOMCAT allows the user to control a series of key experimental settings, affecting the image quality of the resulting CT data [34, 35]. The effects of these experimental settings must be systematically assessed in order to achieve optimal experimental conditions for the specific purpose and required image quality of a study.

In a preliminary experiment, the cementum tissue of specimen l56 was mounted using the convention described above, and imaged using X-ray PPCI through SR CT for a range of different experimental settings. Four key experimental settings (X-ray energy, exposure time, number of X-ray projections, and sample-to-detector distance) were individually varied according to Table 2, while all other experimental settings were fixed at an X-ray energy of 20 keV, a voxel size of 0.66 μm and an exposure time of 150 ms for 1501 projections, at a sample-to-detector distance of 14.00 mm, which corresponds to a similar effective X-ray propagation distance of 13.99 mm [35] due to the parallel X-ray beam geometry at TOMCAT. The effects of changing experimental settings on image quality of cementum increments were characterised by the signal-to-noise ratio (SNR) and the contrast-to-noise ratio (CNR) as figure of merits. SNR quantifies the level of the image signal relative to the background noise. CNR is a useful measure for assessing image contrast between distinct structures, such as dark/light cementum increments.

Image quality measures for each experimental setting were calculated for 10 reconstructed μCT slices, representing the same regions of the tooth root of l56 in each scan (Fig 3). SNR was calculated as the ratio between the mean greyscale value (representing tissue

**Table 2. Image quality assessments of PPCI SR CT images using different experimental settings.**

| Scan name | Energy (keV) | Exposure time (ms) | Number of projections | Sample-to-detector distance (mm) | Mean SNR | Mean CNR |
|---|---|---|---|---|---|---|
| SO20 | 19 | 150 | 1501 | 14 | 187.9 | 50.7 |
| SO21 | 20 | 150 | 1501 | 14 | 123.8 | 56.9 |
| SO22 | 21 | 150 | 1501 | 14 | 112.9 | 60.8 |
| SO23 | 22 | 150 | 1501 | 14 | 82.6 | 35.5 |
| SO24 | 26 | 150 | 1501 | 14 | 72.0 | 24.5 |
| SO25 | 20 | 100 | 1501 | 14 | 118.3 | 50.7 |
| SO26 | 20 | 125 | 1501 | 14 | 124.0 | 55.4 |
| SO27 | 20 | 300 | 1501 | 14 | 177.8 | 72.3 |
| SO28 | 20 | 150 | 3001 | 14 | 245.4 | 74.5 |
| SO29 | 20 | 150 | 4501 | 14 | 252.7 | 81.5 |
| SO30 | 20 | 150 | 6001 | 14 | 300.5 | 84.9 |
| SO31 | 20 | 150 | 1501 | 16 | 142.2 | 59.0 |
| SO32 | 20 | 150 | 1501 | 20 | 152.9 | 68.9 |
| SO33 | 20 | 150 | 1501 | 28 | 185.6 | 75.2 |
| SO34 | 20 | 150 | 1501 | 60 | 179.6 | 77.8 |
| SO35 | 20 | 150 | 1501 | 100 | 247.4 | 54.9 |

SNR = signal-to-noise ratio; CNR = contrast-to-noise ratio.

density) for a 150-pixel × 150-pixel ROI of cementum ($\overline{g_c}$) and the standard deviation of a 150-pixel × 150-pixel sample of background (i.e. air) ($\sigma_b$) in each slice:

$$SNR = \frac{\overline{g_c}}{\sigma_b}. \tag{1}$$

CNR was calculated as the difference of the mean greyscale values of the same ROIs of cementum ($\overline{g_c}$) and air ($\overline{g_b}$), respectively, divided by the pooled standard deviation, following the Pythagorean Theorem of Statistics $Var(X \pm Y) = Var(X) + Var(Y)$ or $\sigma^2(X \pm Y) = \sigma^2(X) + \sigma^2(Y)$ for independent random variables $X$ and $Y$, with $Var$ and $\sigma$ denoting the variance and standard deviation, respectively):

$$CNR = \frac{\overline{g_c} - \overline{g_b}}{[(\sigma_c^2 + \sigma_b^2)/2]^{1/2}}, \tag{2}$$

where $\sigma_c$ and $\sigma_b$ represent the standard deviations of cementum and background, respectively. Mean values of SNR and CNR were calculated from the values for these 10 slices and compared between all experimental settings (Fig 4 and Table 2).

Following this preliminary study, an X-ray energy of 20 keV and a sample-to-detector distance of 14 mm was chosen to scan three ROIs within the coronal third of the mesial root of each tooth, overlapping longitudinally and covering a volume of approximately 4×1.3×1.3 mm³ (Table 1). This provided sufficient image contrast between cementum increments for further image processing and increment analysis. For each scan, the exposure time was set to 150 ms, and 1501 projections were taken per scan. These settings provided sufficient image quality at scan times that allowed the entire sample of teeth to be imaged during our fixed-time experiment. Phase contrast imaging is more sensitive for small differences in material properties in the hard X-ray range compared to (traditional) X-ray attenuation-based imaging [1, 2, 36]. The phase of each X-ray projection was retrieved through the Paganin single-distance non-iterative phase retrieval algorithm [37], implemented in-house at TOMCAT.

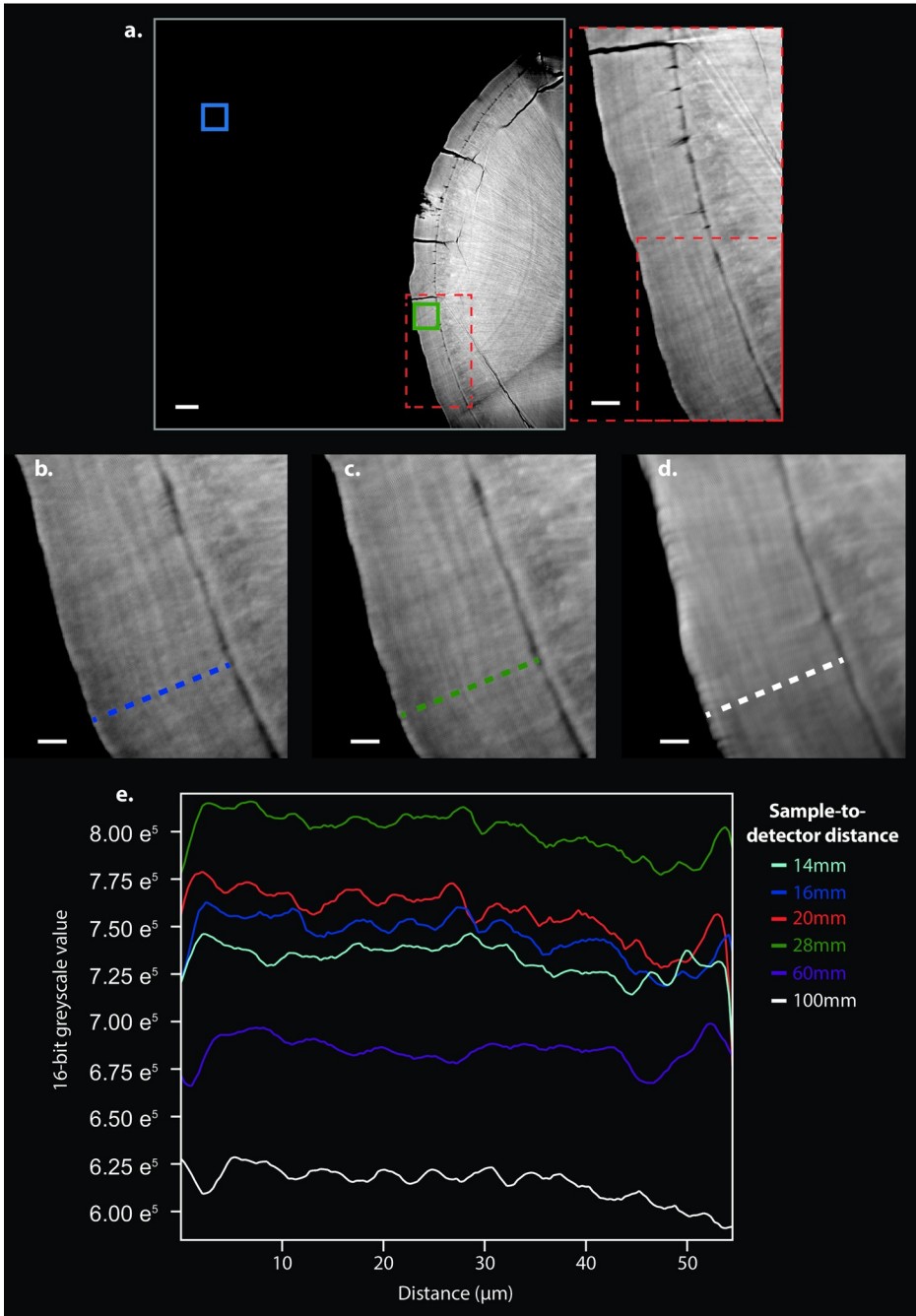

**Fig 3. SR CT scan of a tooth root and regions for signal-to-noise ratio (SNR) and contrast-to-noise ratio (CNR) calculations.** (**a**) One CT slice of the tooth root of *Macaca mulatta* individual l56. Blue box highlights the ROI for evaluation of background signal from which the mean greyscale value ($\overline{g_b}$) and standard deviation of greyscale values ($\sigma_b$) was generated for SNR and CNR calculations (see Eqs (1) and (2)). The green box highlights the sampling area for cementum signal from which the mean greyscale value ($\overline{g_c}$) was generated for SNR and CNR calculations. Dashed red boxes indicate regions highlighted of the detail views. (**b**) Detail from region indicated by dashed red boxes in (**a**) from the dataset acquired at 16 mm sample-to-detector distance. (**c**) Detail from the same region from the dataset acquired at 28 mm sample-to-detector distance. (**d**) Detail from the same region from the dataset acquired at 100 mm sample-to-detector distance. (**e**) Plots of greyscale values along transects indicated by dashed lines in (**b-d**), acquired at different sample-to-detector distances. (**a-d**) White scale bars in (**a**) represent 100 μm in the overview image, 30 μm in the detail view highlighted by the red dashed box, and 10 μm in (**b-d**).

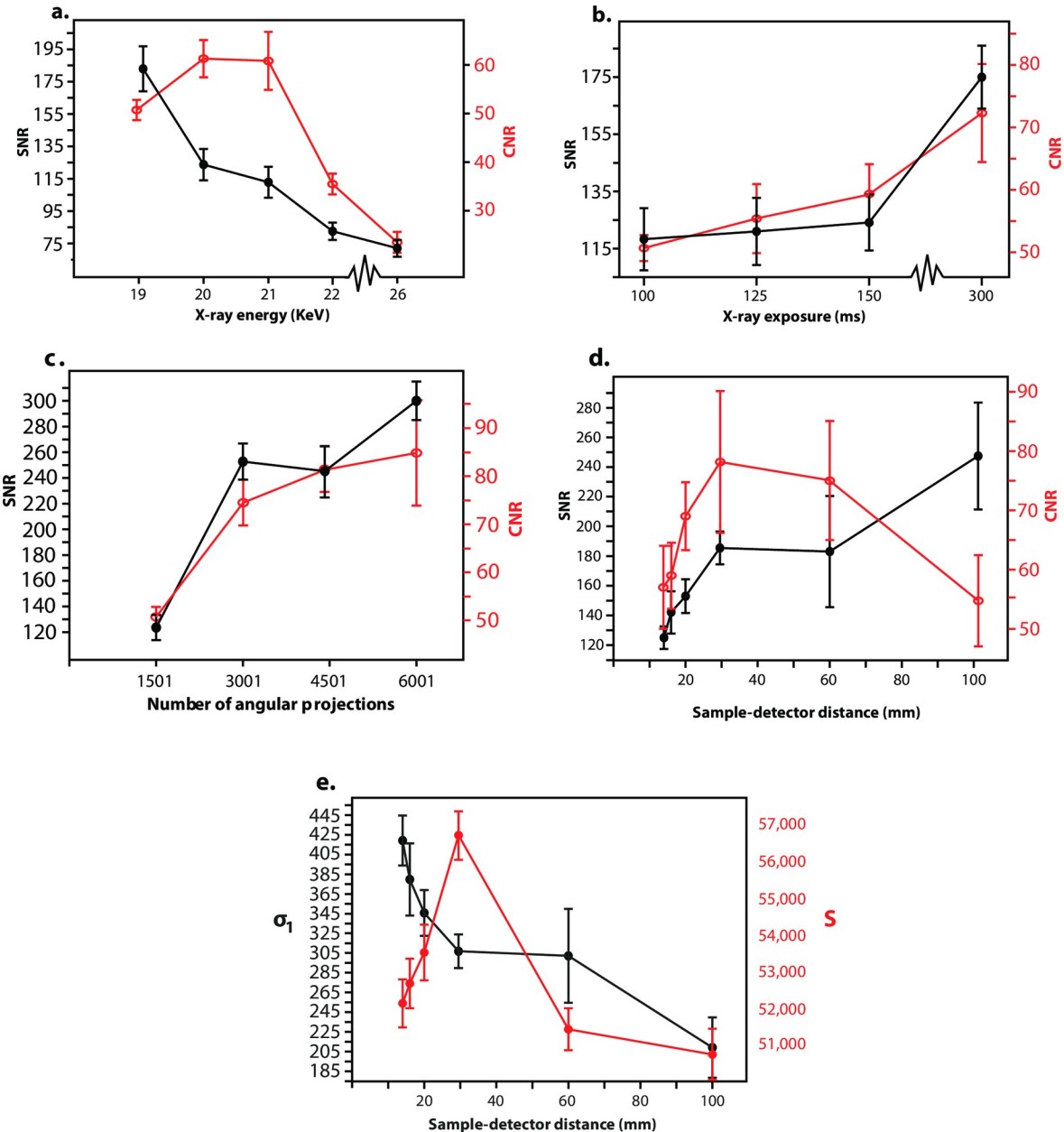

**Fig 4. SNR and CNR for sweep of experimental settings for PPCI through SR CT.** Data shown is from specimen l56. (**a**) SNR (shown in black) and CNR (shown in red) values for different X-ray energies. (**b**) SNR and CNR values for different exposure times. (**c**) SNR and CNR values for different numbers of angular projections. (**d**) SNR and CNR values for different sample-to-detector distances. (**e**) Relationship between the standard deviation of background ($\sigma_b$–shown in black) and the mean of cementum signal ($\overline{g_c}$ –shown in red) with increasing sample-to-detector distance.

The values of the imaginary part of the refractive index $\delta = 3.7 \cdot 10^{-8}$ and the decrement of the real part of the refractive index $\beta = 1.7 \cdot 10^{-10}$, and hence the ratio $\delta/\beta = 218$, were fixed for all scans. The retrieved phase images were reconstructed using an in-house implementation of the Gridrec algorithm [38] at TOMCAT. The resulting CT reconstructions were saved as 16-bit tiff stacks.

## 2.3. Image processing: Straightening and filtering

Image processing of the original reconstruction data is often needed before digital image visualisation, segmentation and quantification. This can involve a wide range of image processing methods, the majority of which are based on the manipulation of 2D pixels and/or 3D voxels using mathematical operations [39]. Here, we applied two principal image processing methods to individual transverse CT slices: straightening and isolation of cementum, and directional filtering of cementum increments. Straightening and isolation involved distinguishing the cementum from the dentine tissue that it surrounds in each slice, removing the dentine data and algorithmically minimising/"straightening" the remaining cementum data along its circumferential axis (Fig 5). This then allowed directional filtering to enhance image contrast in a single direction between straightened cementum increments.

For circumferential structures such as cementum increments (when viewed in CT slices transversely through the tooth root), it is often difficult to apply standard image processing tools and analyses without distorting results, due to complexities in their patterns and boundaries. Hence, 2D straightening algorithms are often applied in order to further analyse the data. We chose to use the 'Straighten' tool of the open source ImageJ/Fiji image analysis software [16]. This tool applies cubic-spline interpolation across a segmented midline of the feature of interest, which is defined by the user. Straightening is then performed using a series of non-linear cubic splines for an arbitrary number of pixels on either side of the midline that can also be determined by the user. Here, we assigned this number on an individual basis for each dataset, based on the (radial) thickness of cementum being imaged in the first/bottom-most slice, to ensure that all cementum, but no dentine, was included in the processed image (see details in Results) (Fig 5A and 5B). This workflow was then repeated in a semi-automated fashion for all CT slices of each dataset following Newham et al. [31], wherein the same midline coordinates and thickness of the segmentation was applied for as many slices as possible from the first slice of a dataset. Due to the change in shape of the root and non-perfect alignment of the sample axis and scanning axis, these midline coordinates had to be adjusted several times through each scan, as the sample changed in size and drifted through the volume. This required manual inspection of the amount of cementum captured by the segmentation through the volume from the first slice, and adjusting the thickness and midline coordinates.

Following straightening, cementum datasets were further processed using directional filtering in order to enhance contrast between increments (see Section 1 in S1 File. 'Image processing by directional filtering') (Fig 5D). Filtering is commonly used to suppress the contribution of unwanted signals such as noise, while preserving and enhancing the targeted signal or image contributions for the analysis in question [40]. We used a custom MATLAB [R2016a; The MathWorks, Inc., Natick, MA, USA] tool called 'SteerGauss' (version 1.0.0.0) developed and made freely available by Lanman [41], in order to employ directional Gaussian filtering of straightened cementum images following Freeman and Adelson [40]. For directional Gaussian filtering, a Gaussian function for a set of 2D ($x,y$) Cartesian coordinates can be prescribed for any arbitrary orientation using a directional derivative operator that interpolates between two 'basic' Gaussian functions, directed at 0° and 90°, respectively. Straightened increments follow similar longitudinal paths, so a steerable filter can be used to select a single orientation of all increments in an image (Fig 5C and 5D). For the current study, the use of a directional Gaussian filter oriented at 90° to the $x$-axis has been shown to substantially enhance image contrast between straightened cementum increments (see Section 1 in S1 File. 'Image processing by directional filtering') (Fig 5).

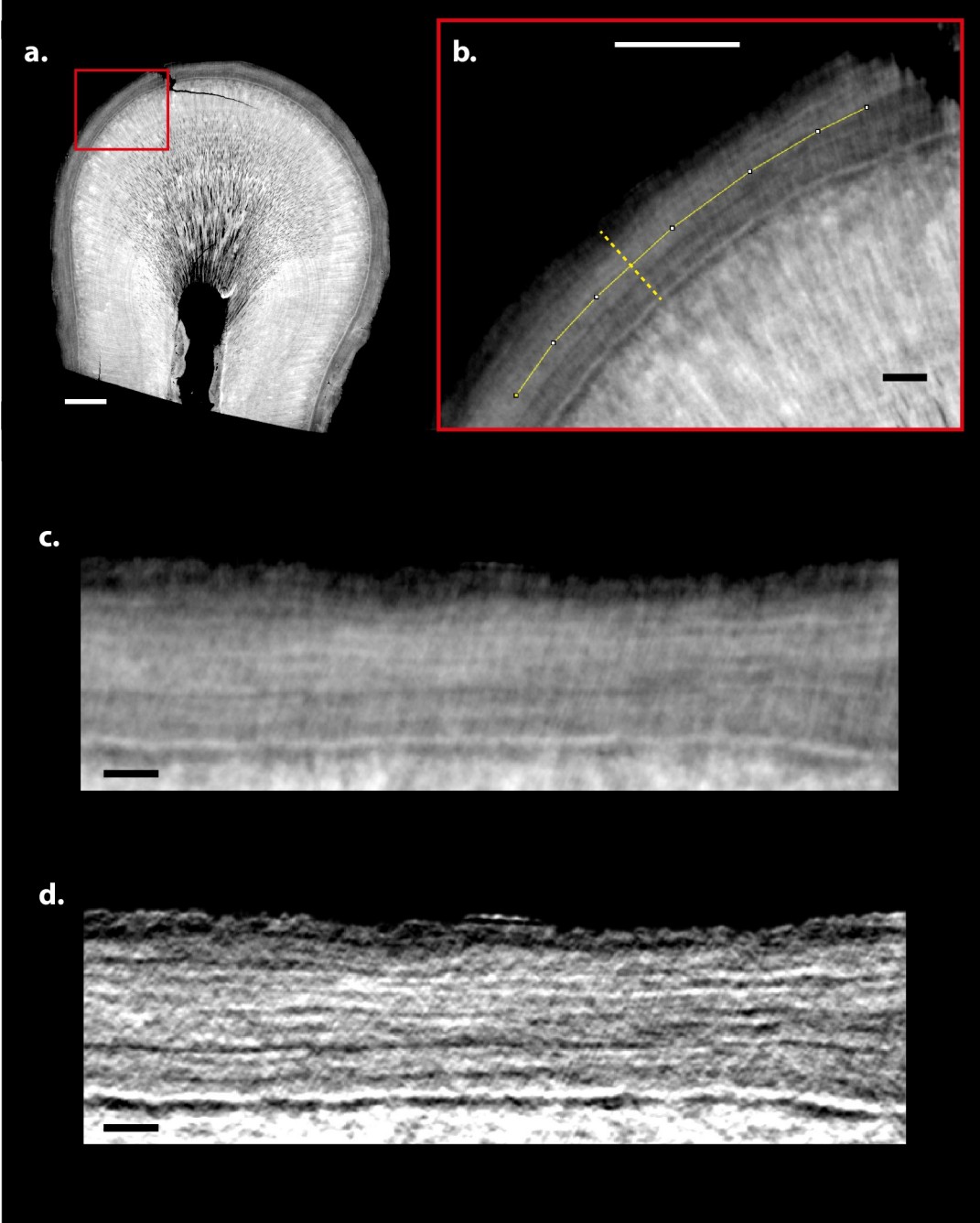

**Fig 5. Image processing of PPCI SR CT images of *Macaca mulatta* cementum.** (**a**) One CT slice of specimen l59. (**b**) Detail of CT slice highlighting circumferential cementum increments (cementum defined by dashed yellow line) with midline shown as solid yellow line. (**c**) Straightened cementum section following the midline highlighted in (**b**). (**d**) Filtered image of (**c**) using a steerable Gaussian filter (see Section 1 in S1 File. 'Image processing by directional filtering'). White scale bar in (**a**) represents 200 μm; and in (**b**) represents 100 μm. Black scale bars in (**c**) and (**d**) represent 30 μm.

## 2.4. Cementum increment counting algorithm

The cementum increment counting algorithm developed here was designed to count cementum increments in a user-independent and semi-automated fashion, further developing the

rationale proposed by DCLA for distinguishing individual increments by using a cut-off point that greyscale peaks/troughs must differ by, in order to be counted as a genuine increment. Our algorithm employs population statistics (mean and standard deviation of greyscale values) to count increments, based on the unique distribution of greyscale values within each individual CT slice (see Section 2 in S1 File 'Robustness testing for increment counting algorithm'). As in DCLA, this method makes use of the average greyscale values along 10 pixel-thick transects through cementum (Fig 2A). Adapting methods used in tribological surface profiling [42, 43], individual transects are separated into five sections of equal length (Fig 2B) (see Section 3 in S1 File 'Splitting of transects through the cementum'). PPCI SR CT datasets of cementum show an overarching reduction in greyscale (density) values from the cemento-dentine boundary to the outer-most cementum increment (Fig 2A and 2B) [31, 32]. Therefore, the use of the mean greyscale value and its standard deviation for the entire transect, as opposed to local values for individual sections of the transect, may preclude the counting of genuine increments towards the outermost increment (as greyscale peaks/troughs are below the mean value), and counting of increments towards the cemento-dentine boundary (as greyscale peaks/troughs are above the mean value). The mean and standard deviation of greyscale values in each section is then calculated (Fig 2B–2D), and light-dark increment pairs are distinguished as peak-trough systems in greyscale that depart from the mean value beyond the local standard deviation in each section (Fig 2C).

Most importantly, this new method for increment counting can be operated in a semi-automated fashion, following an algorithm implemented in the MATLAB statistical environment (see https://doi.org/10.5258/SOTON/D1722 for MATLAB script). In MATLAB, each individual straightened and filtered cementum image is investigated along a series of 1000 transects through the cementum chosen at random using a random number generator (to minimise autocorrelation of increment patterns) (Fig 2A). For each transect, the distance across the transect that the first pixel above zero appears is saved, which gives the radial length of sampled cementum along the transect. Any transect that is less than the lower standard deviation of the saved lengths is then deselected and resampled until all transects fulfil the lower standard deviation of the original sample. Each transect is divided into five sections of equal length (Fig 2B), and a cubic spline ('Smoothing spline' function in MATLAB) is fitted to the greyscale pattern captured within each section, in order to minimise the influence of image noise on peak/trough patterns [44] (Fig 2B). For these five smoothed datasets, their mean greyscale value (red lines in Fig 2) is calculated. An upper 'cut-off' value (green lines in Fig 2) is then determined for each section as its mean greyscale value plus half the standard deviation of its greyscale values, and lower 'cut-off' value (blue lines in Fig 2) as the mean greyscale value minus half of the standard deviation. Two new datasets are then created for each section, the first comprised of only greyscale values above the mean, and the other of values below the mean (Fig 2C). The dataset comprising higher greyscale values thus consists solely of greyscale 'peaks' (local apex in greyscale values), while the dataset comprising lower greyscale values consists solely of greyscale 'troughs' (local nadir in greyscale values) (Fig 2C). The 'Findpeaks' tool (part of the default 'Signal processing toolbox' in MATLAB) is then used to identify peaks and troughs in their respective datasets (following multiplication with -1 to convert troughs into peaks) and calculate their difference from the mean greyscale value of that section. This allows peaks and troughs that extend beyond the top and bottom cut-off values (respectively) for each section to be identified, providing the first stage of estimating increment and increment pair counts (Fig 2B–2D).

Following this first estimate of increment and increment pair counts, further steps are taken within the algorithm to ensure that 'piggy-back' features (secondary peaks/troughs along the ascending/descending limbs of genuine increment peaks and troughs, formed

either due to accessory increments from increment splitting/coalescence, or image noise) do not affect increment counts (Fig 2). No peaks are counted that immediately proceed from the last respective peak; so only one peak is counted for every trough (in Fig 2C peaks i and ii are not counted). Also, no peak/trough system along the transect for which each feature is separated by less than three pixels along the transect (or 1.98 μm) are counted, to ensure that grey scale variations on a small scale do not influence increment counts (Fig 2C peaks i and ii). While this procedure does not necessarily target or discriminate 'accessory' or secondary increments from 'real' circum-annual increments, it remains an objective way of discerning 'real' increments that does not rely on subjective interpretation by the user. Definition of real increment from secondary/accessory increments in-turn relies on user-experience, which is one of the primary elements of cementochronology we are here seeking to minimise.

A final measure is taken to account for increments that are only partly captured inside a neighbouring set of sections along one transect (Fig 2B and 2C). As only the ascending/descending limb of such features would be captured in each section, they may not be detected as a peak/trough in greyscale in either section using the first stage of the increment counting algorithm, which defines peaks or troughs with reference to the two troughs or peaks surrounding them, respectively. A second step is therefore undertaken to distinguish, measure and count these features based on their greyscale values relative to the upper and lower standard deviations of each neighbouring section (described in Section 4 in S1 File 'Accounting for increments split between two neighboring sections'). Once increment pair counts are estimated for the 1000 random transects, the mean and standard deviation are calculated, providing a final estimate of cementum increment pair count for the respective SR CT slice.

The robustness of the proposed algorithm was tested by applying it to a series of digital sine wave patterns of known increment number between five and 30. Random noise at different degrees was applied to these patterns in a controlled manner by increasing their standard deviation along the *y*-axis (Fig 6). Noise was increased incrementally by SNR decrements of 0.1; starting from a SNR of 0.9 and ending at an SNR of 0.1. For each SNR level, increments were counted for 30 sine wave patterns for each count between five and 30. Increment estimates were considered as accurate if the mean estimated count equalled the known increment number to an accuracy of ±0.5. Estimates were considered robust for each count as long as the standard deviation for the 30 counted sine wave patterns was < 1, as values above this may produce estimated increment counts of over 1 year above/below known/expected counts (see Section 2 in S1 File 'Robustness testing for increment counting algorithm').

## 2.5. Application of cementum increment counting algorithm

The increment counting algorithm presented here was used to generate estimates of increment counts for straightened and filtered CT slices for each lower first molar specimen of the 10 *Macaca mulatta* individuals. We applied the cementum increment counting algorithm to 30 CT slices for each individual, representative of highest cementum increment contrast and quality for each individual [6]. Each straightened and filtered CT dataset was examined by eye, in order to find the regions of highest increment contrast and minimum amounts of complexity in increment patterns (i.e. splitting and coalescence of increments). Slices were selected contiguously through these regions until a number of 30 slices had been reached. 1000 transects were plotted through the cementum in each CT slice, and increment pair counts were generated for each transect. The mean increment pair count for all 1000 transects was then used as the estimated increment pair count for the slice, and the mean count of the 30 slices

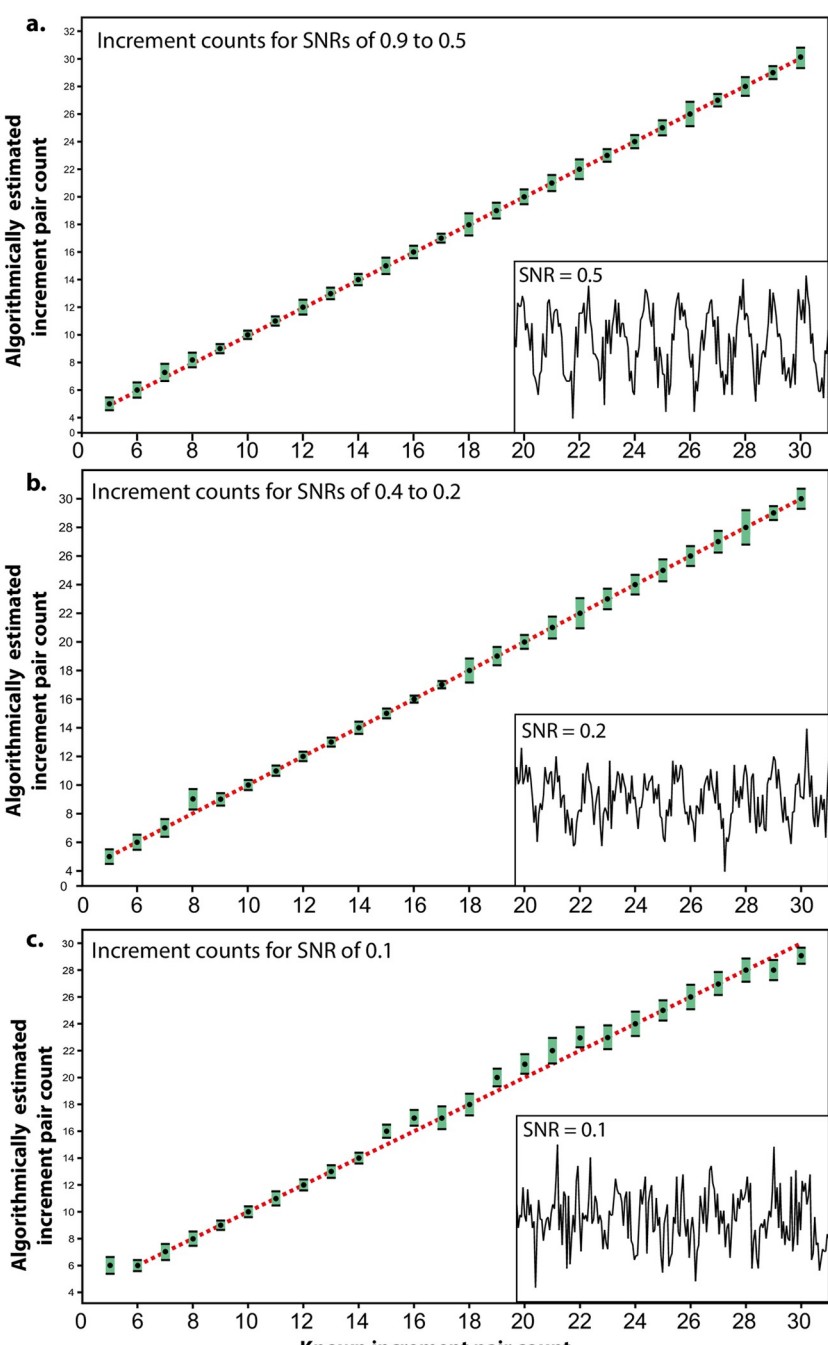

**Fig 6. Robustness tests for algorithmic increment counts.** (**a**) Counts (black circles) and their standard deviations (green boxes) for incremental sine wave patterns of 5 up to 30 increments with signal-to-noise ratios (SNRs) of 0.9–0.5. Inset: box displaying an example of a 10-increment pattern with an SNR of 0.5. (**b**) Counts and their standard deviations for incremental sine wave patterns of 5 up to 30 increments with SNRs of 0.4–0.2. Inset; box displaying an example of a 10-increment pattern with an SNR of 0.2. (**c**) Counts and their standard deviations for incremental sine wave patterns of 5 up to 30 increments with an SNR of 0.1. Inset: box displaying an example of a 10-increment pattern with an SNR of 0.1. Red dashed lines represent 1:1 correlation between estimated and known increment count.

rounded to the nearest integer was defined as the estimated increment pair count for that *Macaca mulatta* individual.

## 3. Results

### 3.1. Optimisation of cementum imaging

When the X-ray energy was changed in isolation, SNR became consistently lower with increasing X-ray energy, whereas CNR peaked at 20 keV, before steadily falling with increasing X-ray energy beyond this point (Fig 4A). SNR and CNR steadily improved with increasing exposure time (Fig 4B). SNR and CNR also improved with increased number of projections, although the relative increase in CNR was marginal between 3001 and 4501 projections (Fig 4C). SNR steadily increased with larger sample-to-detector distances up to 60 mm (Fig 4D). Whereas CNR steadily rose from 14 mm sample-to-detector distance to a peak at 28 mm, it fell between a sample-to-detector distance of 28 mm and 100 mm (Fig 4D).

The image quality of the dataset imaged at 28 mm sample-to-detector distance (Fig 3C) represents an optimum in the trade-off between spatial resolution and contrast for our application. The smoothing inherent in the Paganin phase retrieval algorithm [35] flattens out increment boundaries and reduces noise, while (mean) greyscale differences are retained between light and dark increments to an extent that offers sufficient image contrast to identify individual cementum increments. For datasets created using smaller sample-to-detector distances (14 mm—20 mm), high image contrast resulted between increments, but their boundaries were smoothed and less well defined (case Fig 3B). For sample-to-detector distances above this, the increasing amounts of smoothing diminished the differences in greyscale values between light and dark increments such that by 100 mm sample-to-detector distance, they were difficult to distinguish by eye (case Fig 3D). This can also be shown quantitatively by plotting greyscale values along transects through the same region of cementum in each dataset (Fig 3E) acquired at different sample-to-detector distances.

### 3.2. Cementum imaging results

Cementum was clearly visible in each CT dataset as an incremental tissue wrapping around the dentine of tooth roots and comprising a series of radial increments (Figs 5, 7 and 8). The cementum could be distinguished from the dentine due to its significantly lower mean grey values, and the cemento-dentine boundary was marked by the characteristic tissues of the granular layer of Tomes and the high-density hyaline layer of Hopewell Smith (Fig 7). Individual increments were clearly visible within the cementum and could be followed through the entire dataset, both transversely and longitudinally (Fig 8).

Comparison between CT slices and histological thin-sections of the same regions of cementum (created using the method outlined in Newham et al. [31] and imaged using the method outlined in Section 5 in S1 File 'Thin-section Imaging') suggests that both imaging techniques represent the same cementum increments (Fig 7 and S4 Fig in S1 File). Optical differences between increments in histological data were reflected as grey value differences in CT data. Thick, light increments in histological data corresponded to thick, light increments in CT data relative to thin, dark increments (Fig 7). Volumetric CT data could further be used to help elucidate primary increments from accessory increments in several specimens (Fig 8). Complexities in increment patterns were witnessed intermittently in every *Macaca mulatta* individual, with individual increments splitting and coalescing to create apparent accessory increments. Following Newham et al. [31, 32], individual increments could be mapped through the cementum tissue, and the same primary increments could be plotted through the entire scanned tissue volume (approximately 4 mm$^3$) (Fig 8) across these complexities, and distinguished from the

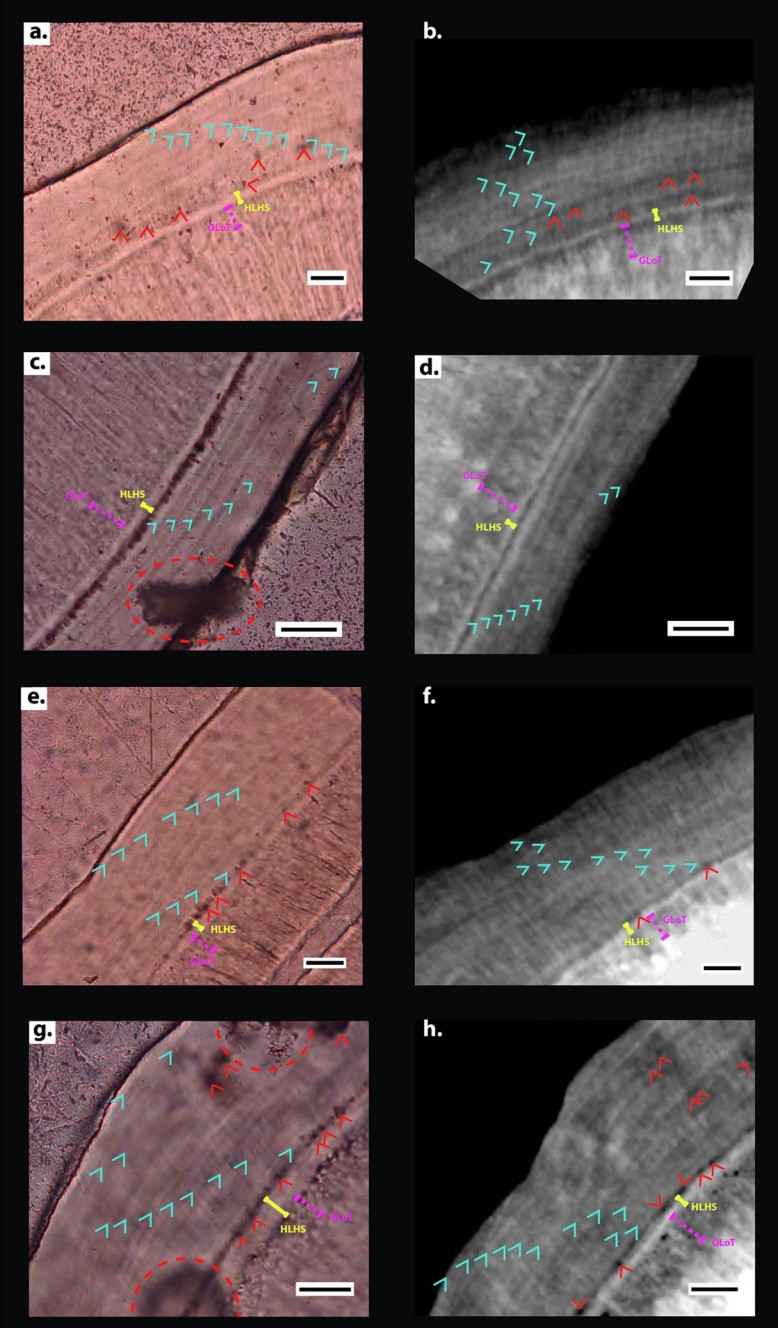

**Fig 7. Comparison between histological and CT data. (a)** Detail of histological thin-section of the k49 specimen displaying 12 light increments indicated by blue arrows. (**b**) Detail of reconstructed CT slice of the same region as (**a**) displaying 11 cementum increments. (**c**) Detail of histological thin-section of the k16 specimen displaying eight light cementum increments. (**d**) Detail of reconstructed CT slice of the same region as (**c**) displaying eight increments. (**e**) Detail of histological thin-section of the l59 specimen displaying 11 light increments. (**f**) Detail of reconstructed CT slice of the same region as (**e**) displaying 11 increments. (**g**) Detail of histological thin-section of the l56 specimen displaying 12 light increments. (**h**) Detail of reconstructed CT slice of the same region as (**g**) displaying 11 increments. See S4 Fig in S1 File for combination of histological/tomographic data for each specimen. Black scale bars represent 30 μm. Yellow whiskers highlight the granular layer of Tomes (labelled GLoT); pink dashed whiskers highlight the hyaline layer of Hopewell Smith (labelled HLHS); and red arrows highlight cellular voids within cellular intrinsic fibre cementum. Red dashed circles highlight surface damage created during thin-section processing.

A semi-automated approach for counting cementum increments in X-ray tomographic data

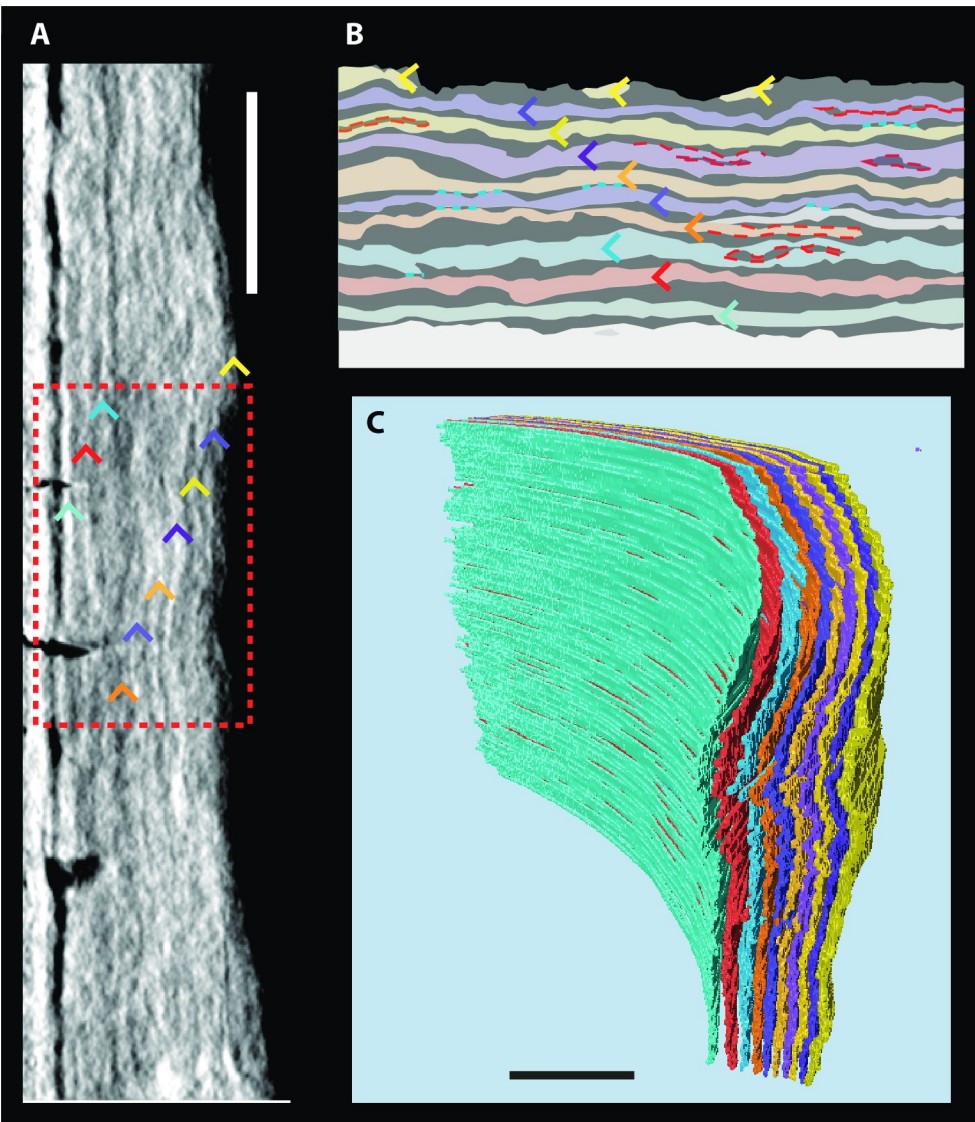

**Fig 8. 3D CT data of cementum increments.** Data shown is from specimen k49. (**a**) Straightened and filtered CT slice displaying 10 increment pairs, highlighted with coloured arrows. (**b**) Schematic of detail from (**a**) highlighted by dashed red box, showing complexities in increment patterns. Increments are given the same colour as their respective arrows in (**a**). Instances of splitting are highlighted with dashed red lines, and instances of coalescence by blue dashed lines. (**c**) 3D model of segmented cementum increment patterns plotted through the coronal third of the root imaged by PPCI through SR CT. Scale bars represent 100 μm.

accessory increments. Therefore, regions that were confounded by splitting and coalescing of these increments could be distinguished and excluded for analysis of increment counts (Fig 8). Also, CIFC, the tissue with the least chronological precision in its increment periodicity [6], could be distinguished from acellular cementum by the presence of cellular voids, and so could be avoided when identifying high-image contrast regions of increments with a circum-annual periodicity (Fig 7). These two factors, aided by the entire coronal (crownward) third of the cementum tissue being imaged, led to the identification of the highest image quality regions of circum-annual cementum increments for each specimen (Fig 7).

PLOS ONE | https://doi.org/10.1371/journal.pone.0249743  November 4, 2021

18 / 26

### 3.3. Validation of cementum increment counting algorithm

Robustness tests for the proposed increment counting algorithm suggest that it is reliable for SNRs down to 0.2 (Fig 6). For each simulated pattern of known increment number, the average value of 30 automated counts was identical to the known count for SNRs between 0.9–0.5. The upper/lower standard deviations of these samples did not exceed one integer above/below the known count (Fig 6). Between SNR levels of 0.5–0.2, average automated counts only differed from known increment number by a value of one in a single sample (with a known increment number of eight). The standard deviations of automated counts exceed one integer above/below the known count for known increment counts of 22 and 28 (Fig 6). SNRs of 0.1 introduced more errors of between one and two in increment count when compared to the known increment number, and the automated count was outside the region of one standard deviations around the known increment number (Fig 6).

When increments were algorithmically counted in our macaque data and compared to expected counts for our sample based on known age, a Spearman's $r$ of 0.77 ($p<0.01$) and Kendall's $\tau$ of 0.71 ($p = 0.004$) suggest significant correlation between semi-automated increment pair counts and expected numbers of cementum increment pairs. The mean of the semi-automated increment pair counts for each *Macaca mulatta* individual either met the minimum or maximum expected count based on their known age or fell in-between the two for every sample, apart from the juvenile individual t46 whose mean estimated count was 0.5 years more than the maximum expected count (Table 1 and Fig 9). Cementum in juvenile individuals, and deposited prior to sexual maturity, has been previously shown to contain more complex incrementation and greater amounts of increment splitting and coalescence than cementum deposited during sexual maturity [45]. Standard deviations of increment pair counts

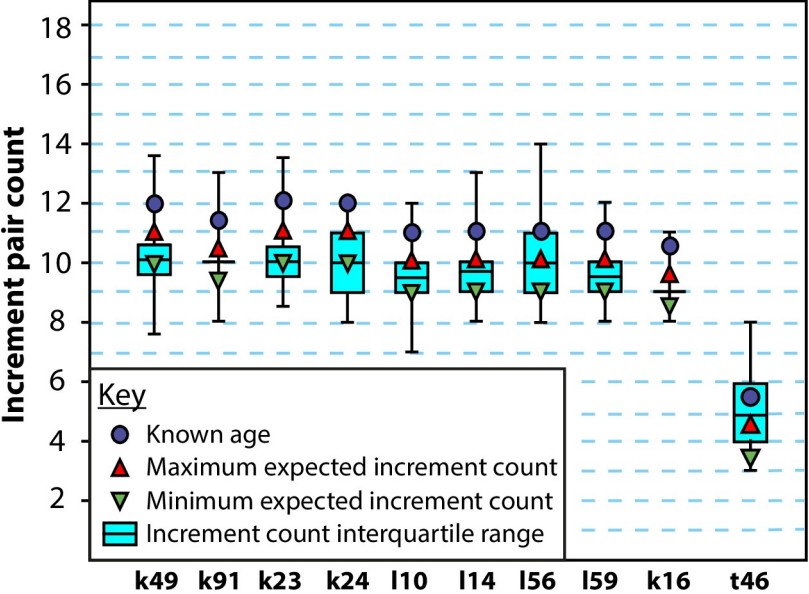

**Fig 9. Comparison between known increment pair counts and those estimated by the proposed cementum increment counting algorithm.** PPCI SR CT data shown are from M1 teeth of ten *Macaca mulatta* individuals of known age (see Table 1). Cyan boxes indicate the interquartile range around the mean estimated increment count, indicated by the thick black line. Whiskers represent the extreme lower and upper estimated counts. Blue circles indicate known age of each individual. Red triangles indicate the maximum and inverted green triangles indicate the minimum expected increment count for each individual based on the approximate age-of-eruption of the M1 at 18 months of age in *Macaca mulatta*.

(average = 0.83) for the 30 individual CT slices examined for each individual did not exceed one for any *Macaca mulatta* individual. This suggests a precision of within one year for estimated increment counts using the proposed cementum increment counting algorithm.

# 4. Discussion

## 4.1. Image quality of SR CT data and optimisation of cementum imaging

The positive relationship observed between both SNR and CNR with increasing exposure time and number of projections has been expected following SR CT imaging of other hard tissues [24, 46]. The opposite relationship seen between SNR and X-ray energy can also be explained by a diminished X-ray absorption with increased X-ray energy due to an exponentially decreased probability of photoelectric interactions between X-rays and the tissue. CNR has a more complex relationship with each experimental setting, with an optimum setting at a different level compared to SNR for each experimental setting. For instance, we located the optimal energy at TOMCAT for cementum increments in terms of CNR at around 20–21 keV, while SNR was highest at 19 KeV and continuously decreased for higher X-ray energies.

The steady increase in SNR with increasing sample-to-detector distance is in agreement with the results of Kitchen et al. [34], but in contrast to the results of Zeller-Plumhoff et al. [35]. Instead, Zeller-Plumhoff et al. [35] found that SNR of PPCI SR CT data of (soft/non-mineralised) muscle tissue steadily decreased when sample-to-detector distance was increased between 30 mm and 60 mm at TOMCAT. The main factors responsible for the increase in SNR with increasing sample-to-detector distance in our study are the steady decrease in the standard deviation of the image background ($\sigma_b$) with increasing sample-to-detector distance versus the peak in mean greyscale value of cementum ($\overline{g_c}$) between 28–60 mm (Fig 4E). The Paganin phase retrieval algorithm acts as a low pass filter, reducing the image noise in the resultant CT reconstructions. This filtering has been enhanced here with increased sample-to-detector and hence propagation distance, as shown by Kitchen et al. [34]. The reason for the different patterns encountered in SNR between the results of Kitchen et al. [34] and those of Zeller-Plumhoff et al. [35] were attributed by Zeller-Plumhoff et al. [35] to be due to different targets in terms of image quality, when considering the optimal ratio of $\delta/\beta$ for the Paganin phase retrieval algorithm. The objective of Kitchen et al. [34], and of our study, was primarily to enhance image contrast within the PPCI SR CT data, whereas Zeller-Plumhoff et al. [35] also considered the sharpness of feature boundaries when optimising $\delta/\beta$. Moreover, the material properties of cementum are different to the soft tissues studied by both Kitchen et al. [34] (lung tissue) and Zeller-Plumhoff et al. [35] (muscle tissue).

It is important to remember that, while our optimisation study was focused on image quality, other factors play important roles for the PPCI of cementum. Firstly, cementum has recently been suggested as an important reservoir of ancient DNA (aDNA) in sub-fossil and archaeological specimens, including early hominins such as Neanderthals [47 and references therein]. It is also well understood that the significant X-ray doses provided by low energy and/or sub-micrometre resolution PPCI is sufficient to denature and destroy aDNA within the scanned ROI [47]. Hence, the optimum experimental settings presented here may well involve an X-ray dose that is too high to be viable for specimens bound for aDNA extraction and study [47].

Further, while increasing the number of angular projections increases image quality, it also increases the total acquisition time of each CT scan. As each synchrotron experiment is awarded a restricted amount of beamtime, optimal experimental settings have to be weighed up against scanning time to image a sufficient sample of specimens. Hence, given the time allocated for our experiment, we retained 1501 projections per CT scan, half of the optimal value

around 3000–4000 in terms of SNR (Fig 4), in order to image our complete sample within the allotted beamtime.

## 4.2. X-ray PPCI versus thin-section imaging for counting cementum increments

The first objective of this study was to optimise PPCI through SR CT for studying cementum increments in 3D tomographic data, as an alternative strategy to destructive thin-sectioning and light microscopy for imaging and counting cementum increments. We have shown here that optimised PPCI strategies can overcome the principal caveats of thin-section imaging: namely the destructive sample preparation process (with the caveat that, in the present study, we isolate both tooth roots from the crown) and the limited 2D view of tissue that is actually 3D in nature, so lacking context for interpreting complexities in increment patterns; and also limited control over which cementum tissue type is imaged (AEFC versus CIFC). The high image quality offered by SR CT, including phase retrieval offered in PPCI, has provided comparable fidelity for counting individual cementum increments to thin-section histological images of the same regions. This is of particular note given that the study sample originates from a captive population with minimal seasonal environmental and nutritional variation. It is understood that the cementum of captive mammals presents lower increment contrast than free-roaming populations [1], and so the identification and accurate counting of circum-annual increments of PPCI data in captive cementum may be another promising element of this technique. This also adds to the body of work finding circum-annual increments in modern humans ([6] and references therein), suggesting that incrementation follows an endogenous (potentially hormone-driven; see Newham et al. [31]) rhythm as opposed to primarily reflecting seasonal dietary differences [4].

The volumetric nature of CT datasets allows navigation through the entire scanned portion of the cementum tissue at an isotropic and sub-micrometre nominal spatial resolution. Individual cementum increments can be followed across regions exhibiting complex cementum patterns, created by splitting and coalescence of increments, and regions of CIFC can be avoided when analysing AEFC. This minimises the potential for inaccurate increment counting. As a non-destructive technique the use of PPCI through SR CT for cementochronology permits the study of cementum in specimens previously beyond the reach of traditional histological analyses that are destructive, including fossils [32] and archaeological specimens [28–30]. As there is no physical thin-sectioning of the tissue involved for CT, images are not affected by tissue preparation artefacts such as scratches on the ground and polished thin section or tissue distortion through the mechanical cutting process (beyond the specific sample preparation steps specific to this study), which can obscure or alter image details on cementum increments [6, 14].

However, our PPCI of cementum through SR CT has also highlighted the sensitivity of cementum image quality to experimental settings. This, alongside other factors described in Section 4.1., suggests that optimisation of experimental settings should be conducted preliminary to every cementochronological PPCI experiment using SR CT, in order to ensure optimised image quality for identifying and counting cementum increments within the constraints given by the broader aims of the particular experiment [30, 47]. Optimal experimental settings are specific to the optics of the synchrotron beamline and the material properties, size and morphology of the specimen, and so should be investigated when any of these factors are changed.

It also became apparent during scanning that micrometre-scale cracks, which are not visible macroscopically, have formed within the cementum tissue (S5 Fig in S1 File), most probably due the interaction of the X-rays with residual water left within the teeth and/or related effects

due to this interaction [47]. Although this damage could not be seen macroscopically, comparison with the lack of observations of any damage in archaeological [30] and fossil [32] teeth confirms that the preparation of 'fresh' teeth was suboptimal for SR CT scanning, as residual water was retained. Further, it cannot be confirmed whether microcracks occurred solely during the scanning process, or started due to demineralisation during fixation in paraformaldehyde. This should be examined in future studies via microscopic examination of the cementum of fixed teeth. Moreover, further optimisation of the preparation procedure for fresh teeth is still needed, that ensures both minimal risk of biological contamination and minimal damage to the tissue during CT scanning. A short-term discoloration of specimens was also noted, with a slight pink hue created within the scanned portion of tooth roots immediately following scanning. This discoloration lasted typically six-eight hours, before the original colour returned.

### 4.3. Cementum increment counting algorithm

The second objective of this study was to provide a validated, semi-automated and robust algorithm for user-independent counting of cementum increments. The manual counting of cementum increments amongst a restricted number of thin-sections per tooth plays a central role in the usual user-dependent approach for counting cementum increments. This subjectivity has led to a wide range of different accuracies and precisions reported for increment counts and their correlation with known age in animal and human samples. Both accuracy and precision in estimated increment counts correlate with the experience of the researcher when compared to known age (i.e. expected increment count) in validation studies [6].

Our algorithm offers a new method for objectively counting cementum increments in a user-independent and semi-automated fashion. This substantially decreases the subjectivity and propensity for human error involved in increment counting (see Section 6 in S1 File. 'Testing the effects of variability in straightening on increment counting'). Within the same selected sample of straightened, isolated and filtered PPCI SR CT slices, our algorithm requires no further human input for counting cementum increments and will estimate the same increment count regardless of the experience of the researcher. The accuracy and precision of this algorithm has been validated here for both simulated data and our experimental sample of *Macaca mulatta* cementum (in both 8-bit and 16-bit data; see Section 1 in S1 File). It could also be further assessed in the same quantitative manner with other PPCI SR CT cementum data from animals of known age. Such assessment will allow for further optimisation of our algorithm and tailoring for a wide range of PPCI SR CT cementum data.

Finally, although we state the advantages of PPCI over traditional thin-section histological imaging here, the validation of our application on thin-section data of cementum from animals of known age may afford its application for thin-section images. If found to be an accurate method for counting thin-section increments, implementation of our algorithm for thin-sections has potential as an important tool for validating the accuracy of counts estimated by-eye, or even discounting the need for counting increments by-eye completely.

### 5. Conclusion

In conclusion, we have undertaken a first systematic experimental study on cementum increment counting for non-fossilised 'fresh' dental tissue, based on a comparison between optimised PPCI through SR CT and thin-section histological imaging. Comparison between these two imaging techniques has shown that PPCI SR CT data can provide sufficient spatial resolution and image contrast to reproduce individual growth increments in the cementum tissue (as well as accessory increments that can be followed through their own trajectory and

discounted, thanks to the 3D nature of CT data). CT reconstructions are of sufficient quality to count increments semi-automatically using image processing, by defining them as peaks and troughs in greyscale values along transects through the cementum. We have implemented this semi-automated method of increment counting as part of a novel workflow of image processing (cementum isolation, straightening and filtering) and analysis (application of a purpose-built increment counting algorithm). This may help future studies to overcome the central caveat facing current studies of cementum increments: the subjectivity inherent in counting increments by eye that depends on the individual researcher. The combination of non-destructive imaging and objective increment counting may open up a new range of specimens, samples and studies not suitable for destructive thin-section analysis, and help to exploit the potential of cementum as a record of life history for archaeology, anthropology, forensic science and palaeontology.

## Supporting information

**S1 File. Contains all supporting tables and figures.**
(DOCX)

## Acknowledgments

We acknowledge the Paul Scherrer Institute, Villigen, Switzerland for provision of synchrotron radiation beamtime at the TOMCAT beamline of the SLS (Experiment 20151391) and would like to thank Iwan Jerjen, Mark Mavrogordato, Orestis Katsamenis, Sharif Ahmed, Christianne Fernee, Juan Núñez, and Priscilla Bayle for their assistance during our beamtime.

## Author Contributions

**Conceptualization:** Elis Newham.

**Data curation:** Elis Newham, Ian J. Corfe, Philipp Schneider.

**Formal analysis:** Elis Newham.

**Funding acquisition:** Elis Newham, Philipp Schneider.

**Investigation:** Elis Newham, Kate Robson Brown.

**Methodology:** Elis Newham, Ian J. Corfe, Philipp Schneider.

**Project administration:** Elis Newham, Philipp Schneider.

**Resources:** Philipp Schneider.

**Software:** Elis Newham.

**Supervision:** Pamela G. Gill, Neil J. Gostling, Ian J. Corfe, Philipp Schneider.

**Validation:** Elis Newham, Ian J. Corfe.

**Visualization:** Elis Newham.

**Writing – original draft:** Elis Newham, Pamela G. Gill, Kate Robson Brown, Neil J. Gostling, Ian J. Corfe, Philipp Schneider.

**Writing – review & editing:** Elis Newham, Pamela G. Gill, Kate Robson Brown, Neil J. Gostling, Ian J. Corfe, Philipp Schneider.

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
