## [Decision Letter · Decision Letter 0]

8 Jun 2021

PONE-D-21-09124

A robust, semi-automated approach for counting cementum increments imaged with X-ray computed tomography

PLOS ONE

Dear Dr. Newham,

Thank you for submitting your manuscript to PLOS ONE. After careful consideration, we feel that it has merit but does not fully meet PLOS ONE’s publication criteria as it currently stands. Therefore, we invite you to submit a revised version of the manuscript that addresses the points raised during the review process.

The manuscript by Newham and coauthors presents an automated method of counting annual increments in acellular cement from microtomographic volumes acquired in synchrotron light. The authors use tooth samples from macaques grown in captivity.

Both reviewers are very positive about the importance of the contribution, but there are several points (especially highlighted by reviewer 2) that need to be clarified before the manuscript can be published.

I have little to add to the extensive and detailed review of reviewer 2, and I suggest the authors take into consideration the concerns raised by reviewer 2.

As a synchrotron user working with fossil, modern exfoliated, and archaeological origin teeth, I never experienced damages like the ones observed by the authors. I think that the fixing for 10 days in paraformaldehyde (and the following drying) could have been responsible for the observed damages. How the authors are sure that the damages weren’t already present before the SR scan (line 557 ff)?

The length of the scale bars is not always reported in the figure captions (eg. Fig.5). Please provide this information.

In addition, the bibliography of the manuscript is not cited in the text following the rules required by Plos One, there are errors in the bibliography and some works cited are not in the bibliography and others in the bibliography are not in the text.

Generally, the impression that one has when reading the text is that it is extremely specialized while a few changes could make it easier to read for readers not completely familiar with the subject.

The appendix with the MatLab script has not been made available to the editor and the reviewers

We look forward to receiving your revised manuscript.

Kind regards,

Luca Bondioli, M.D.

Academic Editor

PLOS ONE

Journal Requirements:

2. In your manuscript, please provide additional information regarding the specimens used in your study. Ensure that you have reported specimen numbers and complete repository information, including museum name and geographic location.

For more information on PLOS ONE's requirements for paleontology and archaeology research, see https://journals.plos.org/plosone/s/submission-guidelines#loc-paleontology-and-archaeology-research.

"This study was part-funded by a Natural Environmental Research Council/Engineering and Physical Sciences Research Council doctoral candidateship (UK; grant number NE/R009783/1). Funding was also provided by Ginko Investments Ltd (Bristol, UK), and the Academy of Finland. "

We note that you received funding from a commercial source: Ginko Investments Ltd

a)Please state what role the funders took in the study.  If the funders had no role, please state: "The funders had no role in study design, data collection and analysis, decision to publish, or preparation of the manuscript."

Reviewers' comments:

Reviewer's Responses to Questions

**Comments to the Author**

1. Is the manuscript technically sound, and do the data support the conclusions?

Reviewer #1: Yes

Reviewer #2: Yes

2. Has the statistical analysis been performed appropriately and rigorously? 

Reviewer #1: Yes

Reviewer #2: Yes

3. Have the authors made all data underlying the findings in their manuscript fully available?

Reviewer #1: Yes

Reviewer #2: Yes

4. Is the manuscript presented in an intelligible fashion and written in standard English?

Reviewer #1: Yes

Reviewer #2: Yes

5. Review Comments to the Author

Reviewer #1: This is an important contribution to the literature on cementum. It defines the nature of an annual increment in acellular cementum and describes a semi-automated method for making objective quantitatively based counts of annual cementum increments. It is significant that these counts closely match those made with traditional transmitted light microscopy in the same specimens.

The study is based on animals that were raised under laboratory conditions (presumably caged for most of their lives). Since this suggests they were buffered from the external seasonal environment and since the presumption is also that their diet was of a consistent nature throughout their lives, the presence of annual increments in their cementum suggests this is an endogenous rhythm and not one that has been assumed by some to be entirely driven by seasonality and or shifts in dietary quality and/or toughness. Some reflection of this in the discussion might be useful for people to at least ponder the implications.

The age range of the animals in this study is as expected much less than the age range in studies of adult modern human cementum. It seems the accuracy of counts of annual cementum increments decreases with increasing age. This may imply later cementum increments, formed say after twenty of thirty years (way beyond the ages of animals in this study), are of a different nature, perhaps more likely fused together or so much slower in formation rate that they can’t be resolved. It might be a good thing to offer some suggestions about the problems this study may not have been able to address as well as those it clearly does address so well and that are discussed. In this way other potential factors affecting, particularly large cementum increment counts in older humans, are less likely to be swept under the carpet and more likely to be addressed in other ways. I’m particularly happy to see Zander and Hurzeler (1958) making it back into the cementum literature. It is one of very few studies that contains data on rates of cementogenesis in different modern human tooth types over many years.

The materials and methods are very carefully described and left and right ‘m1s’ – better to say M1s because they are permanent teeth – were fixed for 10 days in paraformaldehyde. Can the authors present some evidence that there was no demineralisation from the surface inwards as a result of this that might have introduced (or altered) a gradient in the grey scale values e.g. in Figure 2?

Reviewer #2: I would like to congratulate the authors for their very valuable work. I have provided an annotated pdf, and another pdf with some more general comments, to hopeful help the authors further improving their manuscript, and making it easily reachable for a non-specialist audience. I encourage the authors to address my comments.

And I urge them to be careful about the comments about the micro-cracks, see my comments.

6. PLOS authors have the option to publish the peer review history of their article (what does this mean?). If published, this will include your full peer review and any attached files.

Reviewer #1: No

Reviewer #2: No

---

## [Author Response · Author response to Decision Letter 0]

28 Jul 2021

Response to the Academic Editor:

We thank the editor for their positive review of our article and offer for revision and resubmission. We concede that comparison with the results of similar studies on archaeological and fossil teeth suggest that the damage observed in our experiment was most likely due to suboptimal preparation of teeth prior to scanning. We have adjusted our discussion of this accordingly, to minimise any inference that synchrotron imaging itself is necessarily destructive.

We have revised our citation format and reference list to both conform to that of PLoS ONE, and amend inconsistencies in the reference list. We have also revised our scale bar description and scaling in all figures. Finally, following the comments of the editor and reviewers, we have aimed to make the narrative more broad in relation to cementum biology and synchrotron radiation-based X-ray imaging. 

We now include the statement “No permits were required for the described study, which complied with all relevant regulations” to conform to PLoS ONE’s reporting requirements.

We also now explicitly state what funding from Ginko investments was used for (Line 665: “Funding was also provided by Ginko Investments Ltd (Bristol, UK) for purchasing external hard drives and funding travel to the Swiss Light Source, and the Academy of Finland. And on line 673: ”The funders had no role in study design, data collection and analysis, decision to publish, or preparation of the manuscript.” We also amend our competing interest statement as: “The authors report no conflicts of interest. The private funding received from Ginko Investments Ltd does not alter our adherence to PLOS ONE policies on sharing data and materials.” 

We hope that the changes and revisions to our manuscript are satisfactory.

Response to reviewer 1:

Reviewer #1: This is an important contribution to the literature on cementum. It defines the nature of an annual increment in acellular cementum and describes a semi-automated method for making objective quantitatively based counts of annual cementum increments. It is significant that these counts closely match those made with traditional transmitted light microscopy in the same specimens.

We thank the reviewer for their considered and positive review.

The study is based on animals that were raised under laboratory conditions (presumably caged for most of their lives). Since this suggests they were buffered from the external seasonal environment and since the presumption is also that their diet was of a consistent nature throughout their lives, the presence of annual increments in their cementum suggests this is an endogenous rhythm and not one that has been assumed by some to be entirely driven by seasonality and or shifts in dietary quality and/or toughness. Some reflection of this in the discussion might be useful for people to at least ponder the implications.

We agree that the above-mentioned factors point to the circum-annual periodicity of cementum incrementation following an endogenous, potentially hormonal rhythm. We have thus altered our discussion to include (Line 567): “This is of particular note given that the study sample originates from a captive population with minimal seasonal environmental and nutritional variation. It is understood that the cementum of captive mammals presents lower increment contrast than free-roaming populations (Klevezal, 1995), and so the identification and accurate counting of circum-annual increments of PPCI data in captive cementum may be another promising element of this technique. This also adds to the body of work finding circum-annual increments in modern humans (Naji et al., 2016 and references therein), suggesting that incrementation follows an endogenous (potentially hormone-driven; see Newham et al., 2020a) rhythm as opposed to primarily reflecting seasonal dietary differences (Liberman, 1993).”

The age range of the animals in this study is as expected much less than the age range in studies of adult modern human cementum. It seems the accuracy of counts of annual cementum increments decreases with increasing age. This may imply later cementum increments, formed say after twenty of thirty years (way beyond the ages of animals in this study), are of a different nature, perhaps more likely fused together or so much slower in formation rate that they can’t be resolved. It might be a good thing to offer some suggestions about the problems this study may not have been able to address as well as those it clearly does address so well and that are discussed. In this way other potential factors affecting, particularly large cementum increment counts in older humans, are less likely to be swept under the carpet and more likely to be addressed in other ways. I’m particularly happy to see Zander and Hurzeler (1958) making it back into the cementum literature. It is one of very few studies that contains data on rates of cementogenesis in different modern human tooth types over many years.

While several validation studies have found exceptional accuracy using cementochronology to age humans living up to and above 90 years of age (Kagerer and Grupe, 2001; Witwer-Backhofen et al., 2004), others have indeed shown a decrease in accuracy with increasing age (Mani-Caplazi et al,, 2019; see Naji et al., 2016 for a review). It is known that the cementum of long living animals including humans shows an increasing increment density (i.e. numer of increments per unit length), with increment count increasing faster than increases in radial width of the cementum. This increase in increment density evidently makes subjective determination of increments more difficult, given a certain spatial resolution. Thus, it may well be that the optimum parameters found for our sample will be suboptimal for others. Several synchrotron beamlines can now readily achieve effective voxel sizes of around 0.3 µm, which may be preferable for distinguishing densely spaced increments in human cementum. As we state (now in more detail; see Line 537) in our discussion, we believe our results indicate that each experiment based on novel material to cementum SR CT imaging should begin with a parameter sweep to achieve the optimum data relative to the goals of the particular experiment, including the biological nature of the sample itself.

The materials and methods are very carefully described and left and right ‘m1s’ – better to say M1s because they are permanent teeth – were fixed for 10 days in paraformaldehyde. Can the authors present some evidence that there was no demineralisation from the surface inwards as a result of this that might have introduced (or altered) a gradient in the grey scale values e.g. in Figure 2?

We cannot conclusively state whether any demineralisation occurred due to fixation in PFA, and this may be one of the factors that led to microscale cracking witnessed during the experiment (Figure S5). Preparation of the sample and its effects on imaging are now discussed in the revised manuscript (Line 598). For future work, we intend on dehydrating all teeth through desiccation prior to scanning.

References:

• Condon K, Charles DK, Cheverud JM, Buikstra JE. Cementum annulation and age determination in Homo sapiens. II. Estimatres and accuracy. AM. J. Phys. Anthropol. 1986;71:321-330.

• Kagerer P, Grupe G. On the validity of individual age-at-death diagnosis by incremental line counts in human dental cementum. Technical considerations. Anthropol. Anz. 2001;59:331-342.

• Mani-Caplazi G, Hotz G, Wittwer-Backofen U, Vach W. Measuring incremental line width and appearance in the tooth cementum of recent and archaeological human teeth to identify irregularities: First insights using a standardized protocol. Int J Paleopathol. 2019;27:24-37.

• Naji S, Colard T, Blondiaux J, Bertrand B, d’Incau E, Bocquet-Appel JP. Cementochronology, to cut or not to cut? Int. J. Paleopathol. 2016;15:113-119.

• Wittwer-Backofen U, Gampe J, Vaupel JW. Tooth cementum annulation for age estimation: results from a large known-age validation study. Am. J. Phys. Anthropol. 2004;123:119-129.

Response to reviewer 2:

Thank you for asking me to review the work by Newham and colleagues.

This piece of research is most valuable and represent a considerable step forward in the field of cementochronology, with potential implications for studying hominin fossils, yet with some limitations that I will highlight later in this review.

I would like to congratulate the authors for the work and encourage them to consider the comments and suggestions I made, after what I will fully support the publication of this work in PlosOne.

I have provided a detailed annotation of the main manuscript (attached pdf). Please find below some more general comments:

We thank the reviewer for their extremely considered and thorough review. We appreciate their positive interpretation of the value of our study, and have taken on board their critiques. We have thus considerably revised the elements raised by the reviewer, including new supplementary study and modification of discussion points that were rightfully highlighted as easily misinterpreted. We have gone through the detailed comments in the reviewers attached commented PDF, and attempted to correct all points accordingly. Please see below for direct replies to comments:

Abstract

‐ see my comment, the claim are currently not supported by any reported statistical results. Please refer to your correlation tests.

We have now changed the claim to “qualitative comparison” and have created a new figure in the Supplement (Figure S3), where histological and tomographic data are combined to provide qualitative evidence for this statement.

‐ After reading in depth the paper and SOM, I understand why there is no highlight about the non‐destructive nature of synchrotron µCT imaging, this is later explained by the destructive protocol used. Yet, for future directions, it would be good to stress that non‐destructive imaging is possible.

We have added a note in the abstract that synchrotron CT imaging is in general non-destructive, given appropriate preparation of the sample prior to scanning. 

Introduction

‐ errors in biblio

We have adjusted errors and the citation style accordingly.

‐ lack of precision regarding location of AEFC and CIFC.

We have added more detail on current knowledge of cementum tissue distribution (see line 79), although it should not be expected to be homogeneous (Naji et al., 2016). 

‐ improve clarify regarding the use of the word "cementum": complex tissue with sub‐types, indicate here clearly what is being studied.

We have clarified that we are specifically interested in imaging and analysing AEFC.

Methods + Results

‐ the protocol used here is highly destructive! The crown was sectioned, and the roots separated. The end of the introduction involves a kind of blunt criticism about the pilot study of Le cabec et al 2019, yet, they did not cut the teeth, and use higher energy to avoid damaging the DNA!

The preparation procedure is indeed more destructive than that of Le Cabec et al. (2019), because one of our principal aims was validating the optimum experimental settings to achieve optimal image quality for PPCI of cementum, regardless of specimen provenance or the broader scientific goals. This meant ensuring that samples were processed to be optimal in shape for PPCI, with minimal “excess” tissue interfering with the X-ray beam outside of the region of interest being scanned. We now implicitly discuss these concerns in the paragraph starting at line 191.

Of course, this procedure is unsuitable for many archaeological/fossil specimens, and different experimental settings are necessary for maximal possible image quality while minimising X-ray dose (if a DNA analysis is planned following PPCI) and accounting for interference by other roots etc. Briefly, our focus was to push the imaging technique for best possible image quality that could be achieved at TOMCAT beamline. We now discuss the other constraints that will effect ‘optimum’ experimental settings for a particular cementum study from Line 539.

We apologise for our seemingly blunt review of Le Cabec et al. (2019), as this was definitively not our intention. We value the work of Cabec and co-workers as one of the founding studies for the synchrotron study of cementum and have revised our wording accordingly (Line 151: “Due to the uniqueness of their archaeological sample, the canines studied in Le Cabec et al. [30] could not be thin-sectioned.”). We also believe that our comparison with the current gold standard of thin-section histology is a necessary prerequisite for understanding the accuracy of PPCI data and how they reflect genuine biological structures, and the effects of taphonomic and/or diagenetic processes on this structure.

‐ Whether the teeth used here are deciduous or permanent is not clear until the discussion. This should be stated at the beginning of the “Materials”.

The teeth used are all permanent teeth, and we apologise if we did not make that sufficiently clear throughout. We have now added reference to this more frequently and from the start of the “Materials” section. 

‐ There are no explanation about how the thin sections were made? Any polishing involved? Slice thickness?

The full thin-section processing procedure is available in Newham et al. (2020). We have also added the slice thickness in our revised manuscript.

‐ SR µCT acquisitions: Why choosing 1501 projections whereas SNR seems best around 3000‐4000 projections?

We used 1501 projections for the principal experiment in order to scan the entire sample within the allotted experimental time. This compromise is now discussed in terms of general PPCI study of cementum in the revised discussion: (Line 551 “Further, while increasing the number of angular projections increases image quality, it also increases the total acquisition time of each CT scan. As each synchrotron experiment is awarded a restricted amount of beam time, optimal experimental setings have to be weighed up against scanning time to image a sufficient sample of specimens. Hence, given the time allocated for our experiment, we retained 1501 projections per CT scan, half of the optimal value around 3000-4000 in terms of SNR (Fig. 4), in order to image our complete sample within the allotted beamtime”.

‐ There is a lack of explanation regarding what has been scanned: where is the FOV/ROI on the root? What size? Which root? Where along the root length? Was the ROI chosen in a standardized manner for all individuals? Explain the criteria. A figure would be helpful.

We have now explained that scans were centered on the coronal third of the anterior root of each tooth analysed. Three regions were scanned, which were overlapping longitudinally and covering a volume of approx. 4�1.3�1.3 mm3. See Line 248 for this explanation.

‐ Figure 2: I think there is a mismatch between the x values in Fig 2b and Fig 2c? (if you track the peaks, and corresponding x‐values).

Many thanks for highlighting this mistake, which is now corrected.

‐ Cite more often your supplementary information.

The supplementary information is now cited in seven more places than the original manuscript.

‐ 16 or 8 bit data? This needs to be clarified and tested as it may have an impact on the results.

The original CT reconstructions were saved as 16-bit files, but the output of the steerable Gaussian filter function was saved as 8-bit files, and the increment counting procedure was developed for 8-bit data.

‐ Figure 3: on the plot, there are too many curves compared to the figure legend? Also rather indicate the resolution

/ pixel size and not the sample‐detector distances, as this is less intuitive for most non‐specialist readers.

The plot and key represent the different sample-to-detector distances examined, as shown in Figure 5.e. We use the actual distance between the sample and the detector, rather than the effective propagation distance that would require further explanation in the manuscript. We believe that the actual distance is an effective concept for the unexperienced reader that requires minimal knowledge or explanation of how the two factors (physical distance and effective pixel size) interact, especially when the effects of this distance are shown in Figure 3. Finally, due to the quasi-parallel nature of the synchrotron X-ray beam, effective propagation distance is nearly identical to sample-to-detector distance (Zeller-Plumhoff et al., 2017)

‐ How do the method deal with second‐order increments if any? In the SOM I see they are ignored, as well as on the figure showing the I and II in light blue. But is it so frequent? I might be good to show a picture, because those structures do exist. They have been identified also by SXRF.

The method discounts increments with image contrast that does not exceed the local standard deviation of the image grey scale (i.e. “piggy-back features” as shown in Figure 5). However, the method does not directly discriminate between “real” circum-annual increments and “secondary” sub-annual increments. This would require subjective definition between increment types in each slice examined, which in-turn relies on user experience and interpretation. Our goal is instead to be as objective as possible, and to minimise the reliance of cementochronology on user experience (see Line 380: “While this procedure does not necessarily target or discriminate “accessory” or secondary increments from “real” circum-annual increments, it remains an objective way of discerning “real” increments that does not rely on subjective interpretation by the user. This in-turn relies on user-experience, which is one of the primary elements of cementochronology we are here seeking to minimise.”. 

We understand from previous studies (Newham et al. 2020a,b) that secondary increments commonly form due to intermittent splitting of circum-annual increments, which we have attempted to overcome by intensively sampling over each examined slice, and over 30 slices per scan.

‐ how were the 30 slices chosen within the stack? Or was the ROI chosen to involve 30 slices only? This is not clear.

We apologise that we did not make that clear enough. The 30 slices were chosen within regions with highest image contrast for increments upon manual examination of each scan. For each region, slices were selected until 30 slices had been chosen (for each scan). We have made this clearer in our revised manuscript accordingly (line 419: “Slices were selected contiguously through these regions until a number of 30 slices had been reached”). 

- any use of thick slices (average of multiple slices to enhance visibility)? Or only native slices?

Native slices were used.

Discussion

Lines 553‐560: + Fig. S3: the scans generated micro‐cracks: this is of concern!!! Maybe the specimen was still fresh and not totally dry? this could have induced the liberation of free radicals, especially with the low energy used.

This claim is actually going to be harmful to the synchrotron community, and a major concern! Why? Because non‐ specialists are going to be comforted in the general belief that synchrotron imaging damages specimens. So it should be made clear here why and how these micro‐cracks occurred, and not let readers believe this will happen every time!

We apologise for this unanticipated interpretation of this discussion. By no means we intended to undermine confidence in synchrotron imaging, and have significantly modified our discussion to highlight that the preparation of the sample is the primary cause for microstructural damage, exemplified by comparison with data from archaeological (Le Cabec et al., 2019) and fossil (Newham et al., 2020a) data, Line 608: “It also became apparent during scanning that micrometre-scale cracks, which are not visible macroscopically, have formed within the cementum tissue (Supplementary Fig. S5), most probably due the interaction of the hard X-rays with residual water left within the teeth and/or related effects due to this interaction [47]. Although this damage could not be seen macroscopically, comparison with the lack of observations of any damage in archaeological [30] and fossil [32] teeth confirms that the preparation of “fresh” teeth was suboptimal for SR CT scanning, as residual water was retained. Further, it cannot be confirmed whether micro-cracks occurred solely during the scanning process, or started due to demineralisation during fixation in paraformaldehyde. This should be examined in future studies via microscopic examination of the cementum of fixed teeth. Moreover, further optimisation of the preparation procedure for fresh teeth is still needed, that ensures both minimal risk of biological contamination and minimal damage to the tissue during CT scanning.”

Supplementary material

Equations 2 and 3: is the second 0 the symbol for degree (0°)? then it should be superscript to 0 or 90, here it seems to have normal size and it thus seems confusing. Or L42, explain this is for 0°, this would help non‐specialists to understand the notation.

This font has now been changed accordingly.

Fig. S2 and others: individual plots should at least have 1 or 2 key words as title otherwise it is impossible what they represent without reading the full caption.

Titles have been added to plots as suggested. Thank you very much for this useful comment.

L66: so why not running this on the 16 bit data? It needs to be tested if running it on 8 bit induces a loss of contrast/details that would not occur when using 16 bit data.

The output of the filtering function is first calculated at double precision (i.e. “float64” between 0 and 1) format, then converted to the chosen Tiff format (here as 8-bit data). As this conversion is proportional to the distribution of float64 values in the original filtered file, conversion to 16-bit data will experience the same (relative) greyscale distribution as that of an 8-bit conversion, although the absolute greyscale variation will be different. However, we have now run a test on a subsample of five straightened datasets converted to both 8-bit and 16-bit data and subjected to our increment counting algorithm. This test shows that both bit depths produce comparable accuracy of estimated increment counts when compared to expected counts for each specimen. ANOVA comparisons between 8-bit and 16-bit datasets for each specimen only suggest a significant difference in increment counts for one single specimen, and Levene’s test show no significant difference in the variation between 8-bit and 16-bit datasets for each specimen. We interpret these results as suggestive that, while a minority of specimens may produce different ranges of absolute increment estimate values using 8/16-bit data, they are still centred on the same increment count and 16-bit data does not provide significantly higher accuracy or precision for increment count estimates than 8-bit data.

L78‐79: I am not sure to understand how you choose by which integer you divide the 1st directional derivative image.

The optimum integer was chosen as the integer that provided the strongest image contrast between circum-annual cementum increments, while preserving as much greyscale contrast as possible. This was based upon qualitative examination of data filtered using each integer value successively, and monitoring when particular elements of the data were lost. While this is a subjective process, it was necessary to determine that as much data of interest (circum-annual increments) was preserved as possible, while minimizing the presence of greyscale noise. 

L104: is not this circular reasoning? the PPC SR data without any processing, thus including background noise? this needs to be clarified.

While filtering aims to reduce background noise (i.e. unwanted signal), over-filtering can also remove potentially important and useful biological contrast (i.e. wanted signal). Thus, comparison between filtered data and original data was important to monitor what information (wanted vs unwanted signal) was being lost when applying increasingly strong filters, and to establish a cut-off where too much “wanted” biological signal was lost. 

L161: why "tribological" in the title??? and L187, L286: I don't understand the use of this word "tribological"? To me, this has to do when 2 surfaces are in contact and involve movement (e.g., tooth occlusal surfaces and food items).

We agree that this term is confusing and have deleted the term throughout the manuscript.

Table S5: typo.

Typo corrected. Thank you very much for the detailed review, which is greatly appreciated.

References:

• Naji S, Colard T, Blondiaux J, Bertrand B, d’Incau E, Bocquet-Appel JP. Cementochronology, to cut or not to cut? Int. J. Paleopathol. 2016;15:113-119.

• Newham E, Gill PG, Brewer P, Benton MJ, Fernandez V, Gostling N J, et al. Reptile-like physiology in Early Jurassic stem-mammals. Nat. Commun. 2020a;11(1):1-13.

• Newham E. Corfe IJ, Brown KR, Gostling NJ, Gill PG, Schneider P. Synchrotron radiation-based X-ray tomography reveals life history in primate cementum incrementation. J. R. Soc. Interface. 2020b;17(172):20200538.

• Zeller-Plumhoff B, Mead JL, Tan D, Roose T, Clough GF, Boardman RP, et al. Soft tissue 3D imaging in the lab through optimised propagation-based phase contrast computed tomography. Opt Express. 2017;25(26):33451-68.

---

## [Decision Letter · Decision Letter 1]

27 Aug 2021

PONE-D-21-09124R1

A robust, semi-automated approach for counting cementum increments imaged with X-ray computed tomography

PLOS ONE

Dear Dr. Newham,

Thank you for submitting your manuscript to PLOS ONE. After careful consideration, we feel that it has merit but does not fully meet PLOS ONE’s publication criteria as it currently stands. Therefore, we invite you to submit a revised version of the manuscript that addresses the points raised during the review process.

The revised version of the manuscript is improved and more readable. Both reviewers agree that the paper must be published. However, reviewer 2 made some interesting comments that I suggest the authors should address for the benefit of the paper. After the requested minor corrections, the paper will be accepted timely.

We look forward to receiving your revised manuscript.

Kind regards,

Luca Bondioli, M.D.

Academic Editor

PLOS ONE

Journal Requirements:

Reviewers' comments:

Reviewer's Responses to Questions

**Comments to the Author**

1. If the authors have adequately addressed your comments raised in a previous round of review and you feel that this manuscript is now acceptable for publication, you may indicate that here to bypass the “Comments to the Author” section, enter your conflict of interest statement in the “Confidential to Editor” section, and submit your "Accept" recommendation.

Reviewer #1: All comments have been addressed

Reviewer #2: (No Response)

2. Is the manuscript technically sound, and do the data support the conclusions?

Reviewer #1: Yes

Reviewer #2: Yes

3. Has the statistical analysis been performed appropriately and rigorously? 

Reviewer #1: Yes

Reviewer #2: Yes

4. Have the authors made all data underlying the findings in their manuscript fully available?

Reviewer #1: Yes

Reviewer #2: Yes

5. Is the manuscript presented in an intelligible fashion and written in standard English?

Reviewer #1: Yes

Reviewer #2: Yes

6. Review Comments to the Author

Reviewer #1: I congratulate the authors on a very well written paper

I am satisfied all the comments and suggestions raised previously have been dealt with

Reviewer #2: I would like to congratulate the authors on their work for revising their manuscript, which has greatly improved. I support the publication of this work after some minor comments have been addressed. I have again annotated in details the pdf (text with changes in red, + figures + your answers to the reviewers), and here is a summary of my comments:

Title:

- An important aspect that I missed during the first round of review: the title needs to contain either "synchrotron" or "phase contrast", since most readers would believe that, with a Skyscan (conventional µCT scanner) and scanning in absorption mode, they could visualize and reliably identify cementum increments, which is not the case. It should be made clear that this is enabled by synchrotron imaging, and more specifically PPC µCT.

Abstract:

- L38: the authors present the difficulties encountered by previous studies facing the identification of primary vs secondary cementum increments. Yet, no difference is made in the present study during increment identification (actually this is not so clear: it is identified then discarded, therefore the method seems to be able to make the difference). I just want to draw your attention on the fact that this would need to be discussed as this has been presented as a weakness/difficulty of previous studies, which can considerably affect the reliability of increments counts.

Main text:

- bias in sample: only juveniles? See my comment in the previous round of reviews. The algorithm may work better indeed on young individuals, but may be challenged on older individual, whose cementum may have undergone remodeling to some extent, or have more packed increments with less clear borders.

- “juvenile cementum”, “adult cementum”: I recommend changing this very misleading phrasing as it actually refers to whether the individual is juvenile or adult (and within this whether this is a young or old adult).

- use the multiply sign instead of the letter “x”.

- typo for ref. 30 and work by Newham et al.

- L589: this sentence is misleading: PPC SI does not allow to see the whole of the cementum at 0.66µm at once. For non-expert readers this would provide wrong hopes about what can be achieved.

- Fig S3: a comment is left by one of the co-authors.

- Fig. S4: very convincing!

I could not find any feedback in the Authors’s answers regarding my question whether the color of the specimen had changed after scanning, and whether this color change was reversible or not? This is important to report and state clearly. Works at the ESRF had been strongly criticized for changes in the color of the enamel (on fossil and archeological teeth) due to the color center effect. Tafforeau (2008) had showed that exposure to UV light of appropriate wavelength (or even day light but it takes longer) enables recovering the initial state of the sample.

7. PLOS authors have the option to publish the peer review history of their article (what does this mean?). If published, this will include your full peer review and any attached files.

Reviewer #1: No

Reviewer #2: No

---

## [Author Response · Author response to Decision Letter 1]

17 Sep 2021

Reviewer #1: I congratulate the authors on a very well written paper

I am satisfied all the comments and suggestions raised previously have been dealt with

We thank the reviewer for their previous comments to improve the manuscript, and positive response to our paper.

Reviewer #2: I would like to congratulate the authors on their work for revising their manuscript, which has greatly improved. I support the publication of this work after some minor comments have been addressed. I have again annotated in details the pdf (text with changes in red, + figures + your answers to the reviewers), and here is a summary of my comments:

We firstly thank the reviewer again for their detailed comments of our revised manuscript, which we have endeavoured to follow and correct accordingly. We also thank them for their positive summary of the revision. 

Title:

- An important aspect that I missed during the first round of review: the title needs to contain either "synchrotron" or "phase contrast", since most readers would believe that, with a Skyscan (conventional µCT scanner) and scanning in absorption mode, they could visualize and reliably identify cementum increments, which is not the case. It should be made clear that this is enabled by synchrotron imaging, and more specifically PPC µCT.

We have now changed the title to: “A robust, semi-automated approach for counting cementum increments imaged with synchrotron X-ray computed tomography”.

Abstract:

- L38: the authors present the difficulties encountered by previous studies facing the identification of primary vs secondary cementum increments. Yet, no difference is made in the present study during increment identification (actually this is not so clear: it is identified then discarded, therefore the method seems to be able to make the difference). I just want to draw your attention on the fact that this would need to be discussed as this has been presented as a weakness/difficulty of previous studies, which can considerably affect the reliability of increments counts.

We have now amended the identified portion of the abstract to read: (line 45) “PPCI allowed the qualitative identification of primary/annual versus intermittent secondary increments formed by splitting/coalescence. A new method for semi-automatic increment counting was then integrated into a purpose-built software package for studying cementum increments, to count increments in regions with minimal complications.”

Main text:

- bias in sample: only juveniles? See my comment in the previous round of reviews. The algorithm may work better indeed on young individuals, but may be challenged on older individual, whose cementum may have undergone remodeling to some extent, or have more packed increments with less clear borders.

All teeth studied were permanent teeth, as macaques replace their m1 teeth at <18 months of age. Only one individual was a juvenile/young adult (t46; 5.5 years at death), and the nine other individuals were adults. However, while our macaque sample does not present a juvenile bias, the absolute lifespans experienced by our sample may provide a low lifespan bias for the validation results of our increment counting technique, when compared to potential results for long-living mammals including humans. As you say, such samples will present a different challenge than our sample for both imaging and analysing cementum increments, and will require their own optimisation and validation procedures for these tasks. This is a major element of our discussion and conclusion, and we look forward to testing these potential differences, and further optimisation procedures to provide as accurate estimates as possible for increment counts using synchrotron CT and image processing. 

- “juvenile cementum”, “adult cementum”: I recommend changing this very misleading phrasing as it actually refers to whether the individual is juvenile or adult (and within this whether this is a young or old adult).

We agree that this terminology is too vague and have modified our new MS to be more specific: (line 515) “Cementum in juvenile individuals, deposited prior to sexual maturity, has been previously shown to contain more complex incrementation and greater amounts of increment splitting and coalescence than cementum deposited during sexual maturity [45].”

- use the multiply sign instead of the letter “x”.

Now changed to multiply sign ” ×”

- typo for ref. 30 and work by Newham et al.

All typos now corrected according to detailed PDF notes of MS. We thank the reviewer for their thorough checking!

- L596: this sentence is misleading: PPC SI does not allow to see the whole of the cementum at 0.66µm at once. For non-expert readers this would provide wrong hopes about what can be achieved.

We have now corrected this mistake, changing the passage to (Line 596): “The volumetric nature of CT datasets allows navigation through the entire scanned portion of the cementum tissue at an isotropic and sub-micrometre nominal spatial resolution.”

- Fig S3: a comment is left by one of the co-authors.

Comment now removed.

- Fig. S4: very convincing!

We thank the reviewer for their suggestion for this figure and its added support.

I could not find any feedback in the Authors’s answers regarding my question whether the color of the specimen had changed after scanning, and whether this color change was reversible or not? This is important to report and state clearly. Works at the ESRF had been strongly criticized for changes in the color of the enamel (on fossil and archeological teeth) due to the color center effect. Tafforeau (2008) had showed that exposure to UV light of appropriate wavelength (or even day light but it takes longer) enables recovering the initial state of the sample.

We apologize for our omission of a response to this query. The samples did turn a slight pink colour in the regions that encountered the X-ray beam. This lasted a few hours but colour returned to normal after a few days. We appreciate the information and advice regarding UV light and will consider this technique for future experiments. This is now noted in the manuscript (Line 631): “A short-term discoloration of specimens was also noted, with a slight pink hue created within the scanned portion of tooth roots immediately following scanning. This discoloration lasted typically six-eight hours, before the original colour returned.”

---

## [Editor Report · Decision Letter 2]

22 Sep 2021

A robust, semi-automated approach for counting cementum increments imaged with synchrotron X-ray computed tomography

PONE-D-21-09124R2

Dear Dr. Newham,

We’re pleased to inform you that your manuscript has been judged scientifically suitable for publication and will be formally accepted for publication once it meets all outstanding technical requirements.

Kind regards,

Luca Bondioli, M.D.

Academic Editor

PLOS ONE
---

## [Editor Report · Acceptance letter]

22 Oct 2021

PONE-D-21-09124R2 

A robust, semi-automated approach for counting cementum increments imaged with synchrotron X-ray computed tomography 

Dear Dr. Newham:

I'm pleased to inform you that your manuscript has been deemed suitable for publication in PLOS ONE. Congratulations! Your manuscript is now with our production department. 

Kind regards, 

on behalf of

Dr. Luca Bondioli 

Academic Editor

PLOS ONE